# Disease modeling and pharmacological rescue of autosomal dominant retinitis pigmentosa associated with *RHO* copy number variation

**Sangeetha Kandoi[1,2], Cassandra Martinez[1,2], Kevin Xu Chen[2], Miika Mehine[3], L Vinod K Reddy[1,2], Brian C Mansfield[4], Jacque L Duncan[1], Deepak A Lamba[1,2,5]\***

[1]Department of Ophthalmology, University of California, San Francisco, San Francisco, United States; [2]Eli and Edythe Broad Center of Regeneration Medicine & Stem Cell Research University of California, San Francisco, San Francisco, United States; [3]Blueprint Genetics, Bethesda, United States; [4]Section on Cellular Differentiation, Division of Translational Medicine, Eunice Kennedy Shriver National Institute of Child Health and Human Development, National Institutes of Health, Bethesda, United States; [5]Immunology and Regenerative Medicine, Genentech, South San Francisco, United States

**\*For correspondence:**
lamba.deepak@gene.com

**Abstract** Retinitis pigmentosa (RP), a heterogenous group of inherited retinal disorder, causes slow progressive vision loss with no effective treatments available. Mutations in the rhodopsin gene (*RHO*) account for ~25% cases of autosomal dominant RP (adRP). In this study, we describe the disease characteristics of the first-ever reported mono-allelic copy number variation (CNV) in *RHO* as a novel cause of adRP. We (a) show advanced retinal degeneration in a male patient (68 years of age) harboring four transcriptionally active intact copies of rhodopsin, (b) recapitulated the clinical phenotypes using retinal organoids, and (c) assessed the utilization of a small molecule, Photoregulin3 (PR3), as a clinically viable strategy to target and modify disease progression in RP patients associated with *RHO*-CNV. Patient retinal organoids showed photoreceptors dysgenesis, with rod photoreceptors displaying stunted outer segments with occasional elongated cilia-like projections (microscopy); increased *RHO* mRNA expression (quantitative real-time PCR [qRT-PCR] and bulk RNA sequencing); and elevated levels and mislocalization of rhodopsin protein (RHO) within the cell body of rod photoreceptors (western blotting and immunohistochemistry) over the extended (300 days) culture time period when compared against control organoids. Lastly, we utilized PR3 to target *NR2E3*, an upstream regulator of *RHO*, to alter *RHO* expression and observed a partial rescue of RHO protein localization from the cell body to the inner/outer segments of rod photoreceptors in patient organoids. These results provide a proof-of-principle for personalized medicine and suggest that *RHO* expression requires precise control. Taken together, this study supports the clinical data indicating that RHO-CNV associated adRP develops as a result of protein overexpression, thereby overloading the photoreceptor post-translational modification machinery.

## eLife assessment

This study presents an **important** finding that implicates a rhodopsin gene duplication in the progression of autosomal dominant retinitis pigmentosa in patients. The authors utilize a retinal organoid model to demonstrate a similar disease phenotype and suggest defects can be ameliorated by using photoregulin. The data supporting the conclusions are **solid**, but there are some

concerns regarding the strength of the phenotype in retinal organoids. This work will be of broad interest to vision researchers.

## Introduction

Retinitis pigmentosa (RP) is a genetically heterogenous group of 'rod-cone' photoreceptor degenerative diseases that are unified by common clinical features characterized by progressive vision loss, commonly starting as night blindness (*O'Neal and Luther, 2023*). RP affects roughly 1 in 3000–5000 individuals and can be inherited as autosomal recessive, autosomal dominant (ad), or X-linked disease (*Chizzolini et al., 2011*). adRP can be caused by mutations in at least 24 different known genes (*RetNet: Summaries, 2023*) among which mutations in rhodopsin gene (*RHO*) account for 25% of total adRP cases (*Meng et al., 2020*). A mutation in the *RHO* gene was the first identified cause of RP due to a single-base substitution at codon 23 (P23H) leading to protein misfolding and triggering the death of the rod photoreceptors (*Dryja et al., 1990*). *RHO* is located on the long arm of chromosome 3 (3q22.1) and drives the expression of a 348 amino acid G protein-coupled receptor (GPCR) with seven transmembrane domains, a luminal N terminus, and a cytoplasmic C terminus. Rhodopsin protein (RHO) is localized in the densely packed disc membrane of the rod photoreceptor outer segments. Currently >150 different rhodopsin mutations have been identified, all contributing through multiple mechanisms with each having distinct consequences on the protein structure and function (*Athanasiou et al., 2018*). Based on the experimentally studied biochemical and cellular characteristics, several mechanisms have been linked with *RHO* mutation to cause photoreceptor degenerations including protein misfolding, retention and instability in endoplasmic reticulum (ER), glycosylation defects, post-Golgi trafficking and outer segment targeting, dimerization deficiency, altered post-translational modifications and reduced stability, disrupted vesicular trafficking and endocytosis, and impaired trafficking, leading to constitutive phototransduction activation or altered transducin interactions (*Newton and Megaw, 2020*).

In a conference proceedings, we reported copy number variations (CNV) in *RHO* as a novel cause of adRP (*Duncan et al., 2019*). The current study presents a unique opportunity to better understand the pathogenic effects of two extra copies of intact wild-type *RHO* on a single allele at 3q22.1 in a male patient (68 years of age) diagnosed with adRP. Transgenic mice overexpressing wild-type *Rho* have previously demonstrated photoreceptor degeneration, however the precise mechanism of degeneration is still unclear (*Olsson et al., 1992*). Although mice have similar genetics to humans, the distribution, subtypes, quantity of retinal cells (especially photoreceptor cells), and the developmental timeline of the retina differ greatly (*Volland et al., 2015*). Therefore, access to human cells and tissues vis-à-vis the retinal organoid model provides a reliable, translational, and clinically relevant system to gain insights in understanding the pathogenic effects of excessive rhodopsin in photoreceptors. Induced pluripotent stem cell (iPSC)-based models have been used to model several retinal degenerations such as RP (*Tucker et al., 2013*; *Giacalone et al., 2019*), Usher's syndrome (*Dulla et al., 2021*), Leber congenital amaurosis (LCA) (*Parfitt et al., 2016*; *Kruczek et al., 2021*), and *CRX*-associated LCA7 (*Chirco et al., 2021*). This report aimed at presenting the late-onset *RHO*-CNV associated with RP expands the spotlight on modeling a novel cause of retinal disease via a 3D 'disease-in-a-dish' platform.

Patient-specific retinal organoids serves as a versatile tool for testing various therapeutic interventions including small molecule drugs which aim at modulating the pathways (*Moore et al., 2020*; *Liu et al., 2021*). Small molecule-based targeted therapeutics have the potential to cross the blood-brain barrier when administered systemically and can be therapeutically titrated. Among these, nuclear receptors are important targets as they have druggable ligand-binding sites. Approximately 15% of approved drugs target at least 48 members of human nuclear receptors superfamily, and 10 among these are orphan (*Zhao et al., 2019*). Numerous orphan nuclear receptors are expressed in the retina, of which rod-specific nuclear receptor subfamily 2 group E member 3 (*NR2E3*) is a direct target of neural retina leucine zipper (*NRL*), the main rod-specifying gene (*Kobayashi et al., 1999*). *NR2E3* is expressed very early in post-mitotic rods and coactivates the transcription of rod-specific genes including *RHO*, with *CRX* and *NRL* (*O'Brien et al., 2004*; *Cheng et al., 2004*; *Mitton et al., 2000*) while suppressing cone genes. Recent studies have identified photoregulins (PR), small molecules that can target NR2E3. These molecules, especially PR3, have been demonstrated to reduce expression

**Table 1.** Clinical characteristics.

| Subject ID (gender) | Age (years) | Age at first Sx (years) | VA OD, OS | Slit-lamp exam | Dilated fundus exam | Genetics |
|---|---|---|---|---|---|---|
| RM (male) | 68 | 48 | 20/40 20/70-1 | Pseudophakic OU | • Mild disc pallor<br>• RPE atrophy adjacent to optic disc and along arcades with RPE hyperplasia<br>• Bone spicules<br>• Mild, non-foveal cystoid macular edema | Chromosome 3q22 duplication rearrangement<br>• Entire RHO gene<br>• 5' regulatory regions<br>• Flanking genes<br>• 48 kb triplicated region embedded within a 188 kb duplication |
| RC (female) | 33 | N/A | 20/20 20/20 | OU: Normal | OU: Normal | No chromosome 3q22 duplication rearrangement |

N/A = not applicable. OD = right eye. OS = left eye. OU = both eyes. RPE = retinal pigmented epithelium. Sx = symptoms. VA = visual acuity.

of rhodopsin and other rod genes and modify disease progression in mouse models of rod photo-receptor mutation-associated RP (; *Nakamura et al., 2016*; *Nakamura et al., 2017*). We tested the hypothesis that PR3 can mitigate the deleterious effects of the *RHO*-CNV on in vitro patient-specific retinal organoids by reducing the expression of RHO and allowing normal cellular processing. Overall, we demonstrated the establishment of a human retinal organoid model of *RHO*-CNV associated with adRP to gain critical insights into the pathogenesis of disease and to provide a vital screening tool for the development of various novel therapies.

## Results
### *RHO*-CNV patient presented clinical features of adRP

The clinical and electroretinogram (ERG) data of one patient and one healthy unaffected daughter of the patient is included (*Table 1*). Complete ophthalmological examination by fundus photography and spectral domain optical coherence tomography (SD-OCT) revealed features of RP (*Hirji, 2023*) including bone spicule-like pigmentation changes, optic disc pallor, and attenuation of retinal blood vessels (*Figure 1A*, *Figure 1—figure supplement 1*), with outer retinal atrophy due to the loss of the photoreceptor layers, sparing the central foveal region (*Figure 1B*). While the central foveal region was relatively spared in this patient, the macular cones that remained were observed to be damaged by chronic edema, and photoreceptor and retinal pigmented epithelium (RPE) atrophy had progressed into the macula. Genetic testing of the proband by next-generation sequencing (NGS) showed a complex duplication rearrangement of chromosome 3q22, which encompassed the entire *RHO* coding sequence, 5' and 3' regulatory regions, and flanking genes. The rearrangement consisted of a 48 kb triplicated region embedded within a 188 kb duplication, resulting in three apparently intact *RHO* genes on one chromosome and a fourth, unaltered *RHO* gene on the homologous chromosome. Within the duplications, *RHO* copies were flanked by *H1FOO*, *IFT122*, *EFCAB12*, and *MBD4* while the inverted triplication of *RHO* was flanked by partially triplicated *IFT122* and *H1FOO* (*Figure 1C*). Thus, the NGS data shows a duplication-inverted triplication-duplication event, in which the triplicated segment is inverted and located between correctly oriented genomic segments (*Figure 1C*). No other additional causative variants of inherited retinal degeneration candidate genes were identified. Whole genome sequencing supported the rearrangement and extra copies of *RHO* which were identified by NGS. The *RHO* copy number variants were not detected in the unaffected daughter of the patient. Clinically collected four-generation pedigree data of the family (*Figure 1—figure supplement 1*) included two male family members affected with RP, indicating male-male transmission, consistent with an autosomal dominant inheritance. The proband's affected father was not examined as he was deceased but had been diagnosed clinically with RP. The clinical findings were consistent with the hypothesis that CNV in the wild-type *RHO* causes adRP.

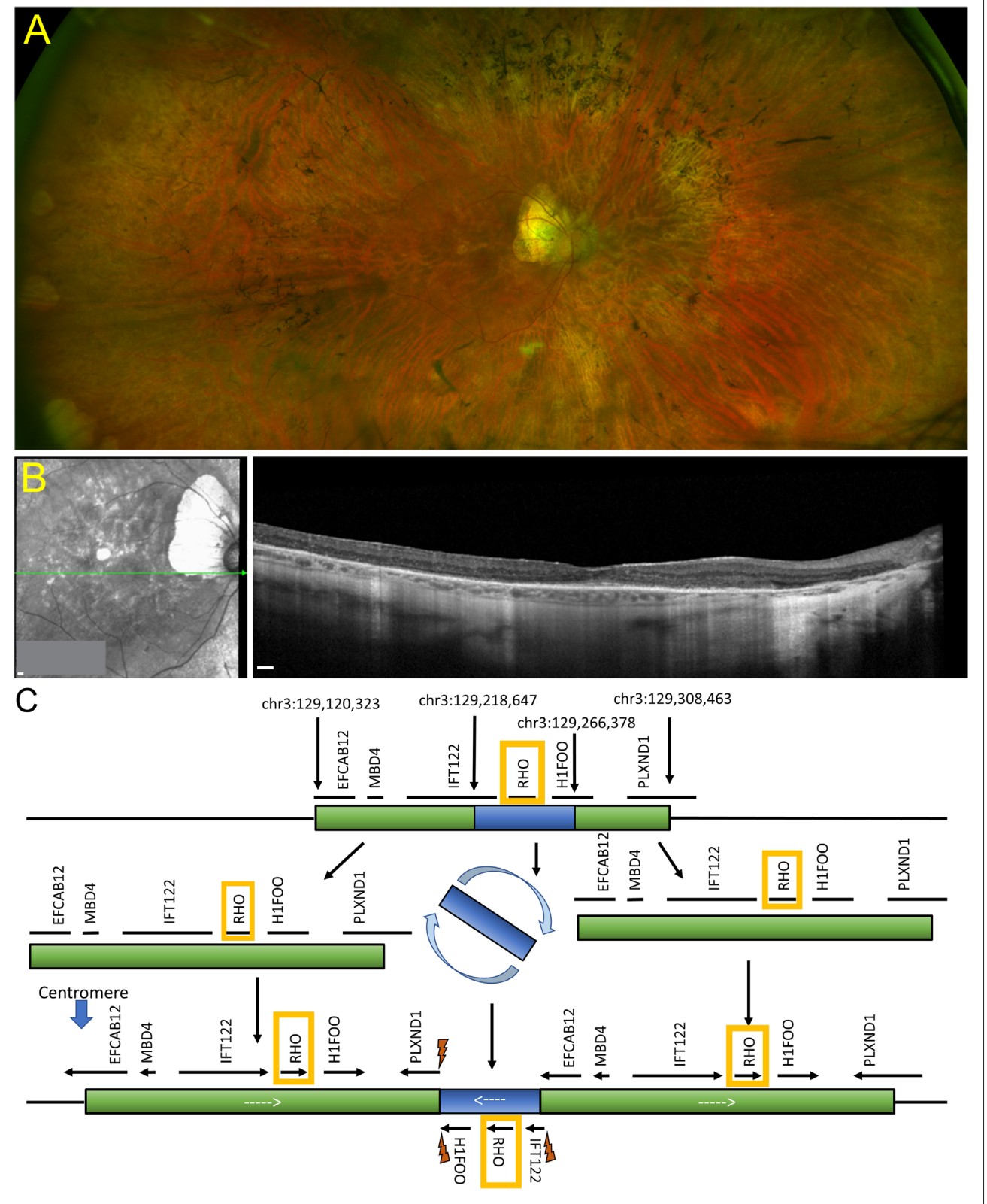

**Figure 1.** Retinal imaging and next-generation sequencing. (**A**) Ophthalmological color fundus examination of a patient clinically diagnosed of *RHO*-CNV displaying bone spicule pigmentary changes. (**B**) SD-OCT image showing extensive loss of the outer retinal layers leaving a small intact island at the fovea. (**C**) Schematic illustration of NGS using a 266-gene retinal dystrophy panel showing a complex chromosome 3q22 duplication rearrangement resulting in an inverted 48 kb triplicated region embedded within a 188 kb duplication forming three apparently intact *RHO* genes on one allele and a

*Figure 1 continued on next page*

*Figure 1 continued*

fourth, unaltered *RHO* on the homologous allele. Lightning bolts represent genomic breakpoints of triplication insertion. *RHO* was the only gene that is fully triplicated. NGS = next-generation sequencing; SD-OCT=spectral domain optical coherence tomography. CNV = copy number variation. Scale bar = 200 µm in A and B.

The online version of this article includes the following figure supplement(s) for figure 1:

**Figure supplement 1.** *RHO*-CNV identification.

## *RHO*-CNV retinal organoids exhibit photoreceptor maturation defects

To assess the effects of *RHO*-CNV in human retinal organoids, we initially reprogrammed the peripheral blood mononuclear cells (PBMNCs) from one patient with four copies of *RHO* (RM) as well as the corresponding familial control (RC) into iPSCs (*Figure 2—figure supplement 1A and B*). The iPSC lines had the colony morphology of tightly packed cells with a high nucleus-to-cytoplasm ratio, a well-defined border typical of stem cells, and expression of pluripotent markers (*Figure 2—figure supplement 1C and D*). We then differentiated the patient and control iPSC lines into retinal organoids by

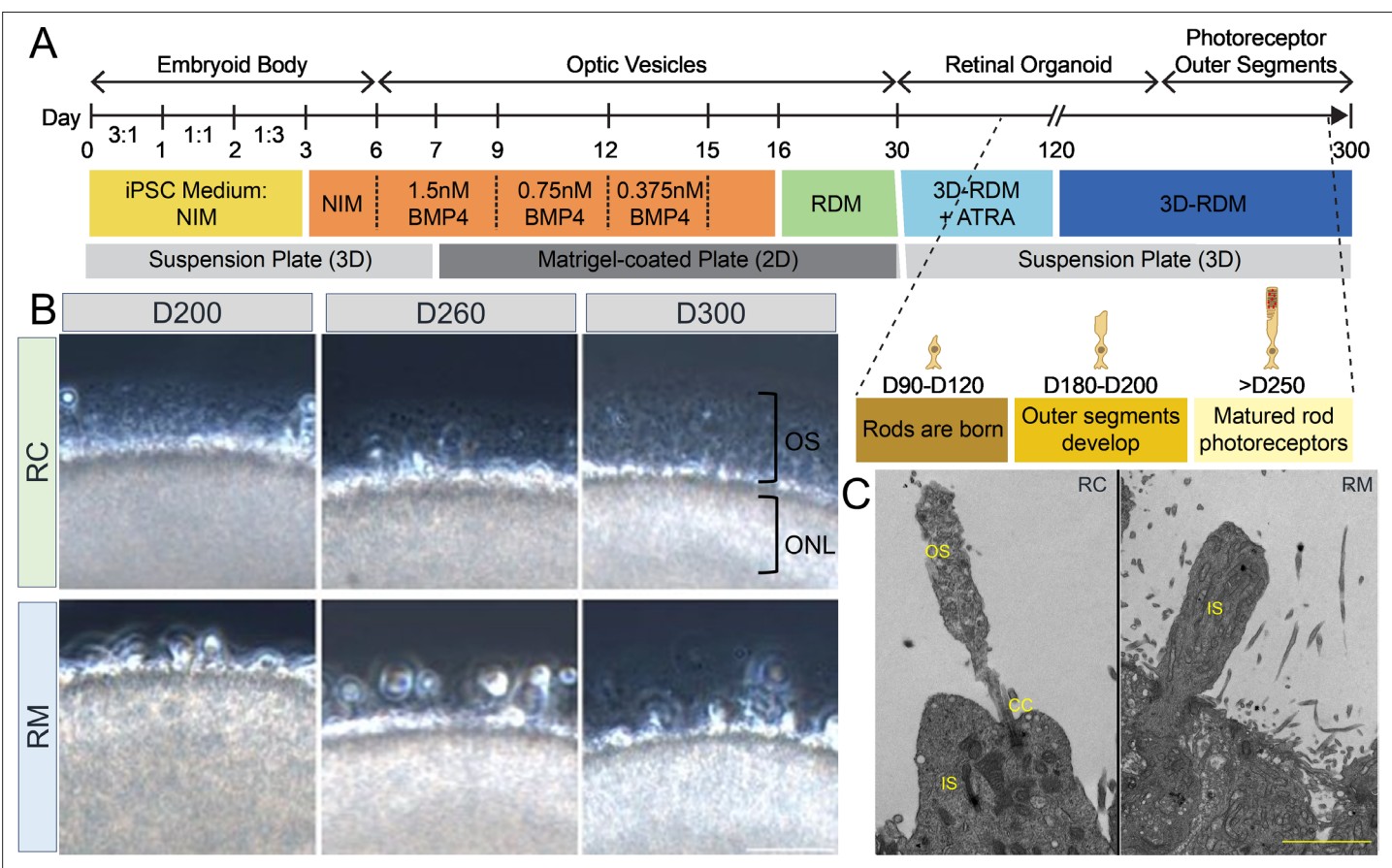

**Figure 2.** *RHO*-CNV disease modeling using induced pluripotent stem cell (iPSC)-derived retinal organoids showed morphological defects. (**A**) Schematic representation showing the timeline of human retinal differentiation and maturation including the birth and development of rod photoreceptors. (**B**) Phase contrast microscopy images showing OS, long hair-like protrusions from ONL of the differentiated photoreceptors present at the apical surface of control (RC) retinal organoids at day 200, 260, and 300 (top). Conversely, retinal organoids from patient (RM) showing shorter protrusions which do not extend progressively over long-term culturing indicating maturation defects (bottom). (**C**) Electron microscopy images showing ultra-magnification of distinct OS, IS, and CC structures of rod photoreceptors in control organoids and the absence of OS in patient organoids. CC = connecting cilium; IS = inner segments; ONL = outer nuclear layer; OS = outer segments. CNV = copy number variation. Scale bar = 50 µm. Raw EM files in attached *Figure 2—source data 1*.

The online version of this article includes the following source data and figure supplement(s) for figure 2:

**Source data 1.** Raw EM files.

**Figure supplement 1.** Patient-specific iPSC reprogramming and retinal organoids.

following our previously published protocols (*Bachu et al., 2022*; *Arthur et al., 2023*), as per the differentiation timeline depicted in *Figure 2A*. Retinal organoids from the *RHO*-CNV patient and the control displayed a well-defined neuroepithelial lamination, indicating the arrangement of photo-receptors on the apical surface of the organoids and a dark central basal region consisting of inner retinal cells by phase contrast microscopy (*Figure 2—figure supplement 1E*). Notably, there were no clear visible anatomical changes in apical-basal retinal cell-type distribution during the early differ-entiation timeframe, when the cone and rod photoreceptors are usually born in 45- to 50-day-old and 90- to 120-day-old human retinal organoids, respectively (data not shown). Over the prolonged differentiation culture timeframe (>200 days), the control retinal organoids displayed long hair-like protrusions which were presumptive inner and outer segments at the apical side of the retinal organ-oids, a critical event indicating the start of photoreceptor maturation. Conversely, the patient retinal organoids showed short initial hair-like protrusions that did not elongate at the extended culture time of 260 and 300 days in culture by light microscopy (*Figure 2B*, *Figure 2—figure supplement 1E*). To validate these observations, we examined the ultrastructure of photoreceptors in the *RHO*-CNV and control organoids using transmission electron microscopy (*Figure 2C*, *Figure 2—figure supple-ment 1F*). Upon assessing the 300-day-old organoids, we observed that while the patient organoids developed connecting cilium and inner segments similar to control organoids, they failed to extend outer segments. This data was consistent across multiple rounds of differentiation from three indepen-dent clones of patient iPSCs. Taken together, these morphological changes suggest that iPSC-retinal organoids of *RHO*-CNV patient demonstrated the survival of photoreceptors with OS dysmorphogen-esis over the time course of 300 days.

## *RHO*-CNV retinal organoids revealed conspicuous defects in rod phototransduction and ciliary transcripts

Control and patient retinal organoids were analyzed at two developmental stages: at rod photore-ceptor birth (D120) and maturation (D300). For each sample, quantitative real-time PCR (qRT-PCR) was carried out by utilizing primers designed to nine key genes that regulate the development and maturation of rod photoreceptors. mRNA levels were analyzed for the expression of genes specific to early pan-photoreceptors (*OTX2, CRX, RCVRN*), early rod photoreceptors (*NRL, NR2E3*), rod-specific phototransduction (*PDE6B, SAG, RHO*), and ciliary (*IFT122*) genes. To equilibrate the data to equiva-lent number of photoreceptors in organoids, we normalized the data to *CRX* expression, a transcrip-tion factor highly enriched in photoreceptors in the retina (*Yamamoto et al., 2020*). The expression of all the target genes were detected at each time-point (D120 and D300) in the retinal organoids (*Figure 3A*). Pan-photoreceptor and early rod marker genes showed similar expression levels with no noticeable variations in RC and RM organoids. In contrast, RHO and SAG were expressed at a higher level in the patient compared to control organoids. There was a significant eightfold increase in the *RHO* levels at D120 and D300 in the patient organoids compared to controls. We also observed a small twofold statistically significant increase in rod/visual arrestin (*SAG*) at D300 time-point in the patient organoids compared to control organoids. Relatedly, a twofold, but non-statistically signifi-cant, change in patient organoids was observed in *IFT122, a* gene partially triplicated in NGS along with *RHO* (*Figure 1C*).

To advance a better molecular understanding on detailed genomic expression and gene regulatory networks, we performed bulk RNA sequencing comparing 300-day-old retinal organoids from patient and control iPSCs (n=3 independent biological replicates). Patient retinal organoids demonstrated upregulated transcriptomic levels of *RHO* (~eightfold) and *SAG* (~fourfold) compared to control organoids, comparable to the differences observed in the qRT-PCR data (*Figure 3B*). Additionally, we also observed increased expression in disc structure support gene including *PRPH*, visual cycle genes *RDH8* and *HCN1*, and a synaptic gene, *PTPRT,* in patient relative to control organoids (*Figure 3B*). Following the gene ontology enrichment analysis (EnrichGO) of significant differentially upregulated genes, we confirmed increases in phototransduction cascades including the visual/light perception and membrane potential/Ca+ signaling pathways in patient organoids compared to control organ-oids. Additionally, increases in genes associated with the synaptic signaling pathway were also detected, suggestive of photoreceptor dysfunction (*Figure 3C*). Pathway enrichment analysis of significantly differentially expressed genes for cellular components category pointed to perturbation in pathways associated with defects in glycosylation especially N-linked glycosylation as well as ER

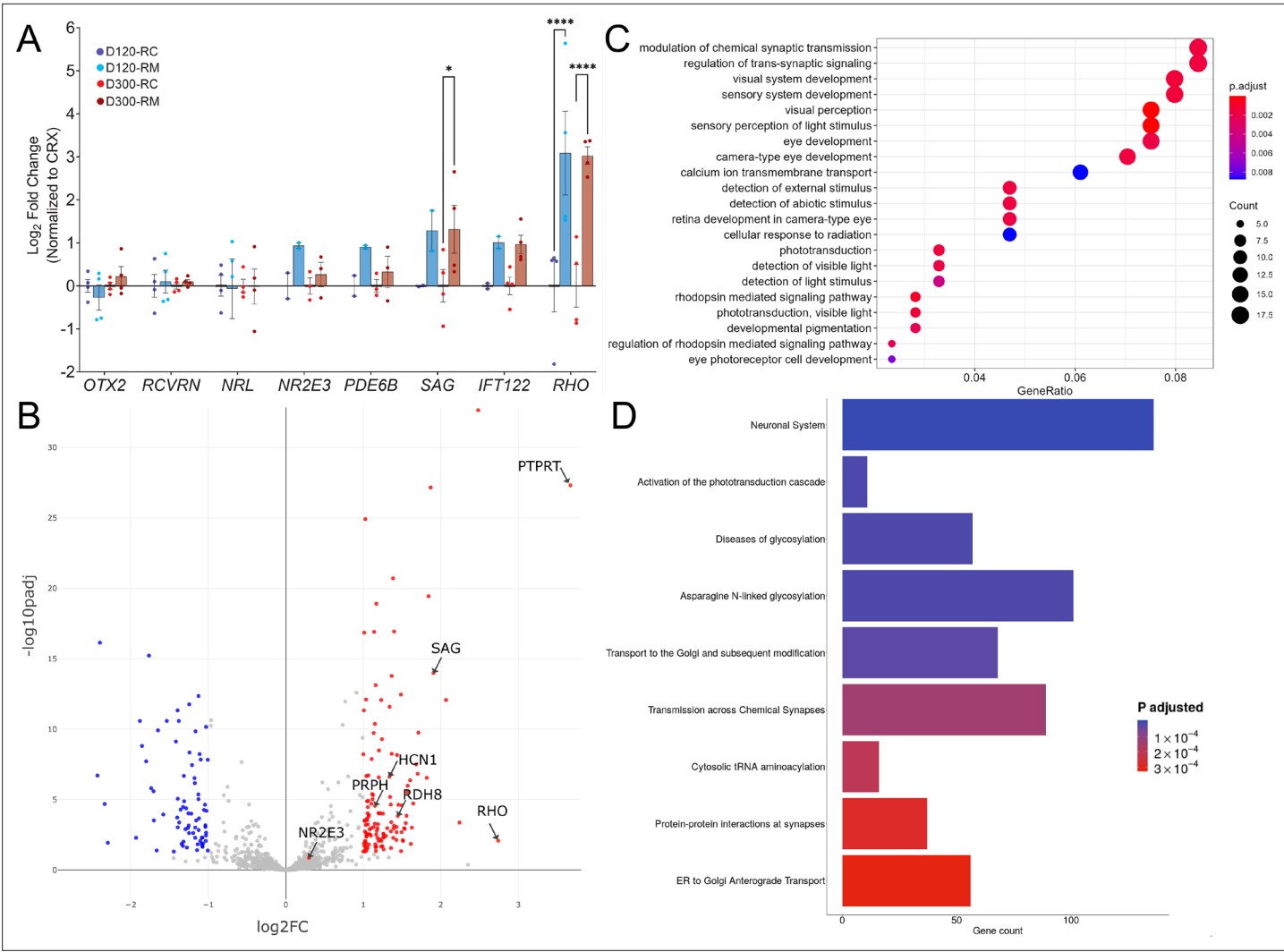

**Figure 3.** Transcriptomic analysis of *RHO*-CNV retinal organoids presented elevated RHO expression. (**A**) Quantitative real-time PCR (qRT-PCR) analysis shows eightfold increase in *RHO* mRNA levels in patient (RM) organoids compared to control (RC) at D120 and D300 of rod differentiation and maturation. No significant change was observed in other photoreceptor genes except for a small twofold increase in rod arrestin (SAG) at D300 time-point. Log2FC = Log2Foldchange. Statistical two-way ANOVA with Fisher's LSD test and 95% confidence interval. *=p<0.05, and ****=p<0.0001. Blue bars, D120; Maroon bars, D300. Error bars represent SEM. (**B**) Volcano plot showing significant differentially expressed genes following bulk RNA sequencing analysis comparing patient to control organoids (>D300). Significantly upregulated genes are highlighted in red and significantly downregulated genes are highlighted in blue (adjusted p<0.01). (**C**) Dot plot showing EnrichGO analysis of biological process on the differentially expressed genes in the bulk RNA sequencing analysis. The size of the dot represents number of differentially expressed genes in the pathway and the X-axis represent the ratio over all genes associated with the pathway. Plot shows a defect in rods and phototransduction-associated pathways as well as synaptic transmission suggesting rod dysfunction. (**D**) Box plot showing the data from pathway enrichment analysis of cellular component category predominantly highlighting the defect in glycosylation and Golgi/endoplasmic reticulum (ER) modification/transport. Colors in the dot and blot plots represent relative significance (calculated p-values in scale). N=3 (RNA sequencing) and N=4 (qRT-PCR) independent experiments and 12–15 organoids per experiment. CNV = copy number variation.

and Golgi transport (*Figure 3D*, *Supplementary file 2*), highlighting potential pathophysiology that may drive *RHO*-CNV-associated RP. Although glycosylation is not required for the rhodopsin biosynthesis, N-linked glycosylation (at N2 and N15), a post-translational modification, is a necessary step for interacting with chaperones during ER transport. This is also an essential step toward incorporation of the heptahelical GPCR rhodopsin in the rod outer segments. Thus, RNA sequencing data suggests that the defects in rhodopsin glycosylation decreased the ability of rhodopsin to exit ER, and lead to an adRP phenotype (*Tam and Moritz, 2009*; *Sung and Tai, 1999*).

## *RHO*-CNV retinal organoids displayed mislocalized and elevated rhodopsin protein levels

Immunofluorescence staining on the cross sections of control and patient retinal organoids was examined for spatial location of the photoreceptors at three time-points (D120, D200, D300). Pan-photoreceptor precursor and progenitor proteins (OTX2, CRX), and rod-specific proteins (NR2E3) were expressed in the apical layer of the organoids in a similar pattern both in the control and patient organoids with equivalent photoreceptors and outer nuclear layer thickness (*Figure 4—figure supplement 1A–C*). These results corroborated our gene expression and transcriptomics data (*Figure 3A and B*) indicating no defects in rod biogenesis within the patient retinal organoids. As the organoids began to mature, the distribution of RHO protein was observed in the outer segments of control organoids

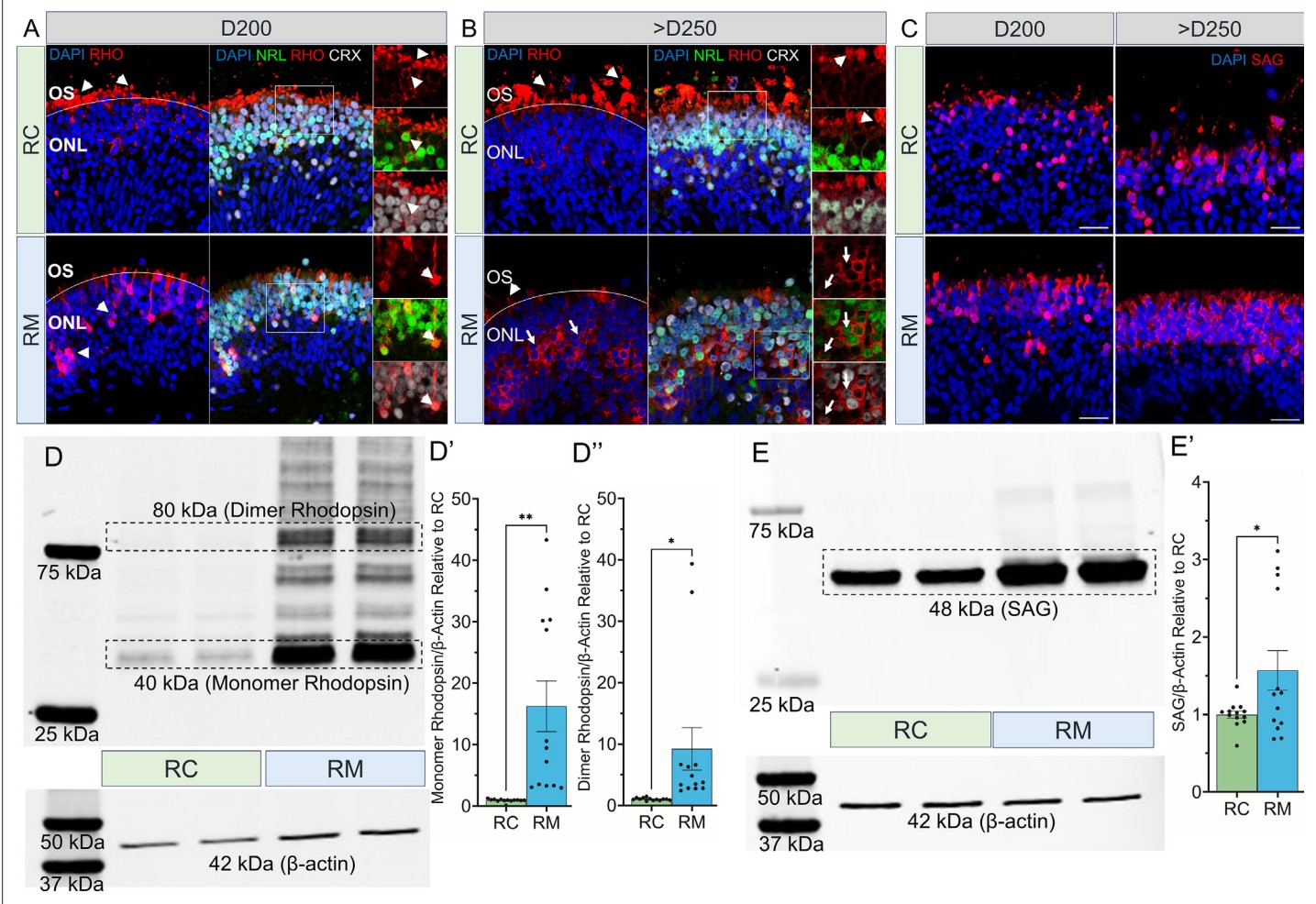

**Figure 4.** Rhodopsin protein mislocalization and increased levels in *RHO*-CNV retinal organoids. (**A, B**) Immunofluorescence staining of RHO, NRL, and CRX displaying the proper RHO localization (arrowheads) in the outer segments of control (RC, top panel) organoids and RHO mislocalization (arrows) in the cell body of photoreceptors within the patient (RM; bottom) organoids at two time-points, D200 and >D250. Occasional inner/outer segments with appropriate RHO localization in the patient (RM) organoids were seen at >D250 time-point (arrowhead). (**C**) SAG expression (red) was also increased in the cell soma of patient organoids (bottom) compared to control organoids (top). DAPI (blue) labels nuclei. OS = outer segment; ONL = outer nuclear layer. Scale bar = 25 μm for A-C. (**D, E**) Western blot probed for RHO and SAG showing increased levels of 40 kDa monomer and 80 kDa dimer, RHO (**D**), and 48 kDa, SAG (**E**) in patient retinal organoids compared to controls. β-Actin was used as a loading control. Densitometric analysis quantifying the relative intensity of monomeric RHO (**D'**), dimeric RHO (**D''**), and SAG (**E'**) in comparisons to the control. Statistical two-tailed unpaired t-test analysis with 95% confidence level. *=p<0.05, and **=p<0.01. N=5 independent experiment and 12–15 organoids per experiment. Error bars represent SEM. Raw blots in *Figure 4—source data 1*. CNV = copy number variation.

The online version of this article includes the following source data and figure supplement(s) for figure 4:

**Source data 1.** Western blot files.

**Figure supplement 1.** Rod biogenesis and development in *RHO*-CNV retinal organoids did not differ from control.

starting at D200, earliest time-point for detection in human retinal organoids using our differentiation protocol. Compared to the control, the patient organoids had mislocalized RHO protein accumulating in the photoreceptor cell soma, at all analyzed time-points (D200 and >D250) (*Figure 4A and B*). The patient organoids rarely showed RHO in IS/OS which were notably consistent with our results discerned by light microscopy (*Figure 2B*, *Figure 2—figure supplement 1E*). Co-staining with NRL and CRX confirmed mislocalized RHO expression in the rod photoreceptor soma within patient organoids. We also evaluated the expression of rod-specific phototransduction protein, SAG, which was increased in the outer nuclear layer of patient retinal organoids (*Figure 4C*). We did not observe any differences in the cone markers, ARR3, BCO, or pan phototransduction marker, RCVRN, when comparing >D250 patient to control organoids (*Figure 4—figure supplement 1A-D*). There was also no difference in apoptosis between the patient and control organoids (*Figure 4—figure supplement 1E*).

To investigate protein expression levels in *RHO*-CNV organoids, we quantitated the level of RHO and SAG by western blot by comparing to control organoids. Patient retinal organoid homogenates displayed a significant 16-fold and 9-fold higher fractions of ~40 kDa monomer and ~80 kDa dimer rhodopsin content respectively in patient organoids relative to controls despite loading equal amounts of protein lysates by western blot (*Figure 4D–D"*). A significant 1.5-fold increase in ~48 kDa SAG, a rhodopsin interacting protein, was also observed (*Figure 4E–E'*). Additionally, we analyzed the ER stress-unfolded protein response (UPR) pathway in the patient organoids and did not detect any differences compared to control (*data not shown*). These findings suggest that *RHO*-CNV organoids actively translated the excessive rhodopsin protein which is not getting transported to the outer segments.

## PR3 treatment attenuates *RHO* expression and partially rescues RHO trafficking in *RHO*-CNV retinal organoids

Since the excess RHO protein production due to the extra *RHO* copies is likely not being processed appropriately, we aimed to test potential therapies that may reduce RHO protein levels in rods. We targeted orphan nuclear receptor, *NR2E3*, using a small molecule, PR3, to assess its effect on *RHO* regulation and expression in the human patient organoid model (*Figure 5A*). PR3 has been previously described to regulate the rod gene expression in Rho$^{P23H}$ mice following a screen for molecules targeting NR2E3, suggesting its potential for the use in the treatment of RP (*Nakamura et al., 2017*). We treated long-term cultured 300-day-old *RHO*-CNV patient retinal organoids with varying concentrations of PR3 (0.1, 0.25, and 0.5 μM) for 1 week and assessed the effects on *RHO* mRNA expression and protein localization. Immunofluorescence staining of PR3-treated organoids displayed a partial rescue of RHO localization (*Figure 5B*). 0.5 μM treatment led to the strongest reduction in RHO with very few rods still expressing the protein. 0.25 μM treatment lead to best outer segment region localization of RHO. None of the organoids showed any evidence of toxicity (by TUNEL, *Figure 5—figure supplement 1A*) post treatment. Following qRT-PCR analysis of PR3-treated organoids compared to vehicle-treated ones, we observed a 4-to-30-fold decrease in *RHO* expression in a dose-dependent manner, along with smaller decreases in other rod-specific genes including *NR2E3*, *GNAT1*, *and PDE6B* (*Figure 5C*). We did not see any significant effects of PR3 on blue-, green- and red-cone opsin genes or protein expression for these protein or cone arrestin (*Figure 5—figure supplement 1B and C*). Upon comparison of PR3-treated patient organoids with control (RC) organoids, *RHO* expression levels in the patient organoids treated with 0.1 and 0.25 μM PR3 were not significantly different from control organoids. However, 0.5 μM PR3 resulted in significantly decreased *RHO* expression, much lower than the *RHO* levels observed in control organoids (*Figure 5D*).

We further carried out bulk RNA sequencing analysis to comprehensively characterize three different groups of organoids, 0.25 μM PR3-treated and vehicle-treated patient organoids and control (RC) organoids from three independent differentiation experiments. Consistent with the qRT-PCR gene expression analysis, the results showed a significant downregulation in *RHO* and other rod phototransduction genes including *SAG* and *GNAT1* between the PR3 and vehicle-treated patient organoids (*Figure 6A*). Additionally, we confirmed that PR3 did not have any adverse effects on cone opsin transcripts in patient organoids. Principal component analysis and normalized read counts analysis of sequenced data for rod (*RHO, SAG, GNAT1, NR2E3*) and other genes demonstrated that the PR3-treated organoids were more alike to control organoids than the vehicle-treated patient organoids

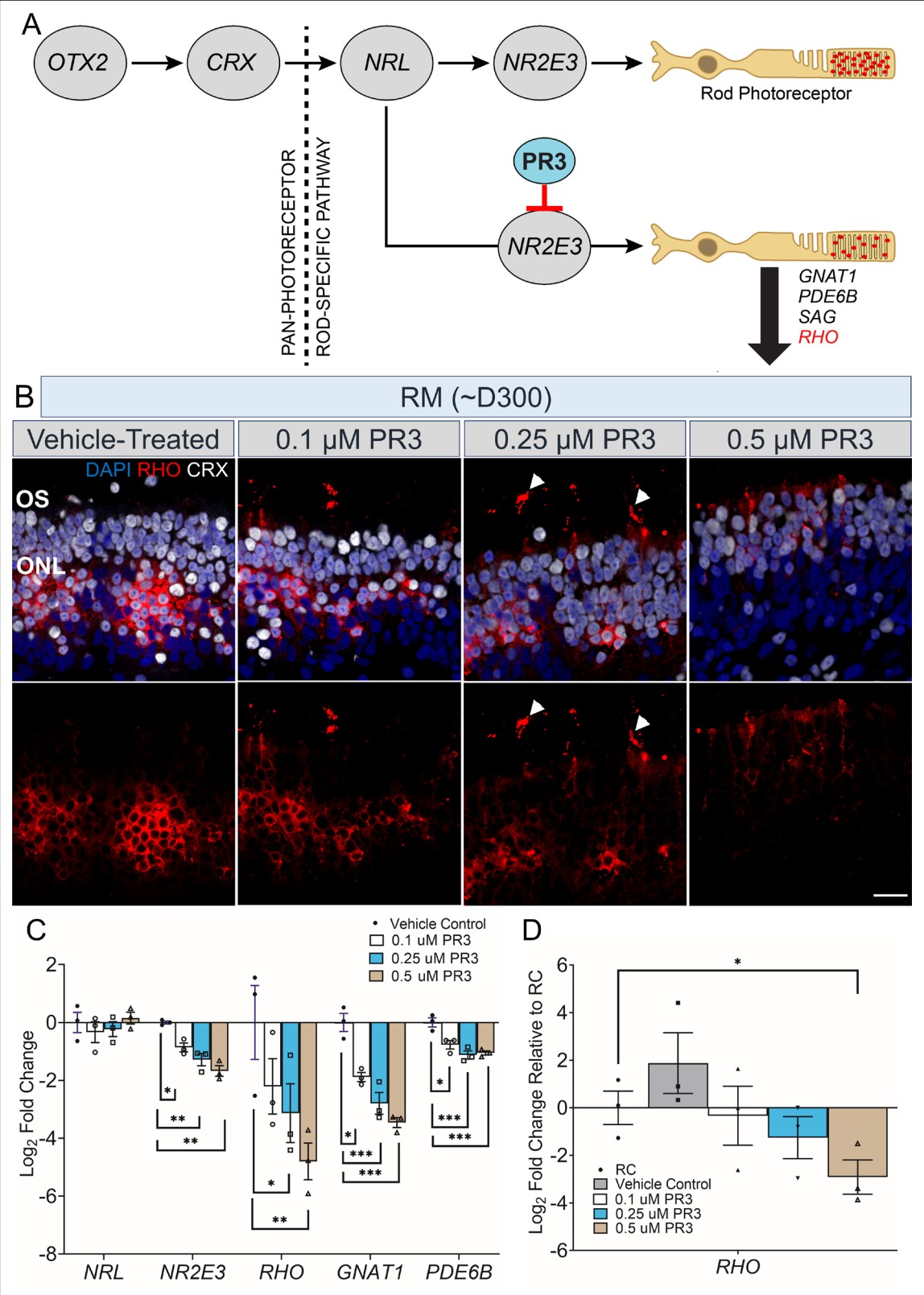

**Figure 5.** Partial rescue of rhodopsin localization and expression levels in Photoregulin3 (PR3)-treated *RHO*-CNV retinal organoids. (**A**) Cartoon illustration showing gene expression during the stepwise development and maturation of rod photoreceptors. Small molecule PR3 is predicted to act on *NR2E3* downregulating the expression of *GNAT1*, *PDE6B*, *SAG*, and *RHO*. (**B**) Immunostaining of ~300-day-old PR3-treated retinal organoid sections from patient (RM) showing the trafficking of RHO protein (arrowheads) toward outer segments at all three doses of PR3, 0.1, 0.25, and 0.5 μM. Retinas

*Figure 5 continued on next page*

*Figure 5 continued*

co-stained for CRX to mark photoreceptors. The most appropriate RHO localization to OS is seen at 0.25 µM PR3 (arrowheads) and much lower overall RHO expression in 0.5 µM PR3. DAPI (blue) labels nuclei. OS = outer segment; ONL = outer nuclear layer. Scale bar = 25 µm. (**C**) Quantitative real-time PCR (qRT-PCR) analysis shows 8- to 30-fold decrease in *RHO* mRNA levels in a dose-dependent manner for PR3-treated patient (RM) organoids. No significant change was observed in *NRL*, but a small decrease was observable in *GNAT1, PDE6B,* and *NR2E3* (one-way ANOVA with Sidak test and 95% confidence interval). (**D**) qRT-PCR analysis showing a comparison of *RHO* mRNA levels in PR3-treated patient organoids to control (RC) organoids (unpaired t-test). Log2FC = Log2Fold change. Error bars represent SEM. *=p<0.05, **=p<0.01 and ***=p<0.001. N=3–4 independent experiments and 12–15 organoids per experiment. CNV, copy number variation.

The online version of this article includes the following figure supplement(s) for figure 5:

**Figure supplement 1.** Photoregulin3 (PR3)-treated *RHO*-CNV retinal organoids do not affect cone opsin expression.

(*Figure 6B and C*, *Figure 6—figure supplement 1A*). Upon KEGG analysis of differentially expressed genes associated with the rod phototransduction pathway between the three groups of organoids, we observed that several rod pathway components which were upregulated in patient organoids compared to control organoids were potentially salvaged following PR3 treatment (*Figure 6D*). EnrichGO analysis of significantly downregulated genes in visual perception/phototransduction pathways confirmed the specific activity of PR3 in retina (*Figure 6E*). Analysis for significantly upregulated genes pointed to changes in cilium organization, axoneme assembly, ER to Golgi transport, and glycosylation. All of these genes are critically required for outer segment formation and protein trafficking (*Figure 6F*, *Figure 6—figure supplement 1B and C*) suggestive of rod photoreceptor maturation recovery. Thus, the data presented strongly suggests that PR3 could potentially rescue rod photoreceptor homeostasis in *RHO*-CNV patients.

## Discussion

To our knowledge, there have been no previous reports of RP associated with multiple copies of wild-type *RHO* in humans. However, retinal degeneration phenotypes have been reported in mice bearing extra copies of *Rho*. A 10–30% increase in rhodopsin expression in mice caused gradual degeneration, while a three- to fivefold increase resulted in severe and rapid photoreceptor loss (*Olsson et al., 1992*). More recent studies show that the retinal degeneration phenotype correlates closely to the amount of *RHO* overexpression (*Tan et al., 2001*; *Wen et al., 2009*). The current study reports the clinical retinal phenotype of a patient with *RHO*-CNV, and photoreceptor abnormalities in iPSC-derived retinal organoids from the patient. Using patient derived organoids provides unprecedented access to human patient-specific material to better understand specific disease processes. These studies also have a caveat to slow disease development due to the time involved in retinal maturation in organoid critical for pathological processes to kick in. Another shortcoming of the current study is the lack of comparison of the organoid phenotype to an isogenic line. While the creation of isogenic lines and comparing them with disease organoid models is an ideal approach, carrying out large deletions by CRISPR-cas9 such as those required to fix the duplicated 188 and 48 kb inverted triplicated region in our *RHO*-CNV patient line is challenging effort. Large CRISPR edits generally have very low efficiency (*Eleveld et al., 2021*; *Canver et al., 2014*). Thus, our study utilized an unaffected first-degree relative as a control.

In this patient, a few surrounding genes are duplicated and partially triplicated as well. Of these, IFT122 is particularly interesting as it has been shown to be part of the ciliary transport. IFT122 has been shown to cause recessive phenotype in dogs and in complete knockout zebrafish model but dominant or overexpression has not been shown to have a phenotype. We did a thorough survey through literature and through BluePrint Genetics database. We did not find any human retinal degeneration cases with variants in IFT122. Interestingly, recessive biallelic IFT122 mutation can cause cranioectodermal dysplasia (Sensenbrenner syndrome). However, none of these patients exhibited retinal dystrophy (*Alazami et al., 2014*). Among the other genes, HIF1OO is an epigenetic modifier gene, MBD4 is involved in DNA repair while PLXND1 is expressed in endothelial cells, and none of these have been implicated in photoreceptor development or disease.

Our data strongly supports the notion that the RP phenotype in the patient is likely due to RHO accumulation and mislocalization in the patient's photoreceptors. Based on studies in animal models, *RHO* mutations could lead to death of rod photoreceptors by several mechanisms, including: (a)

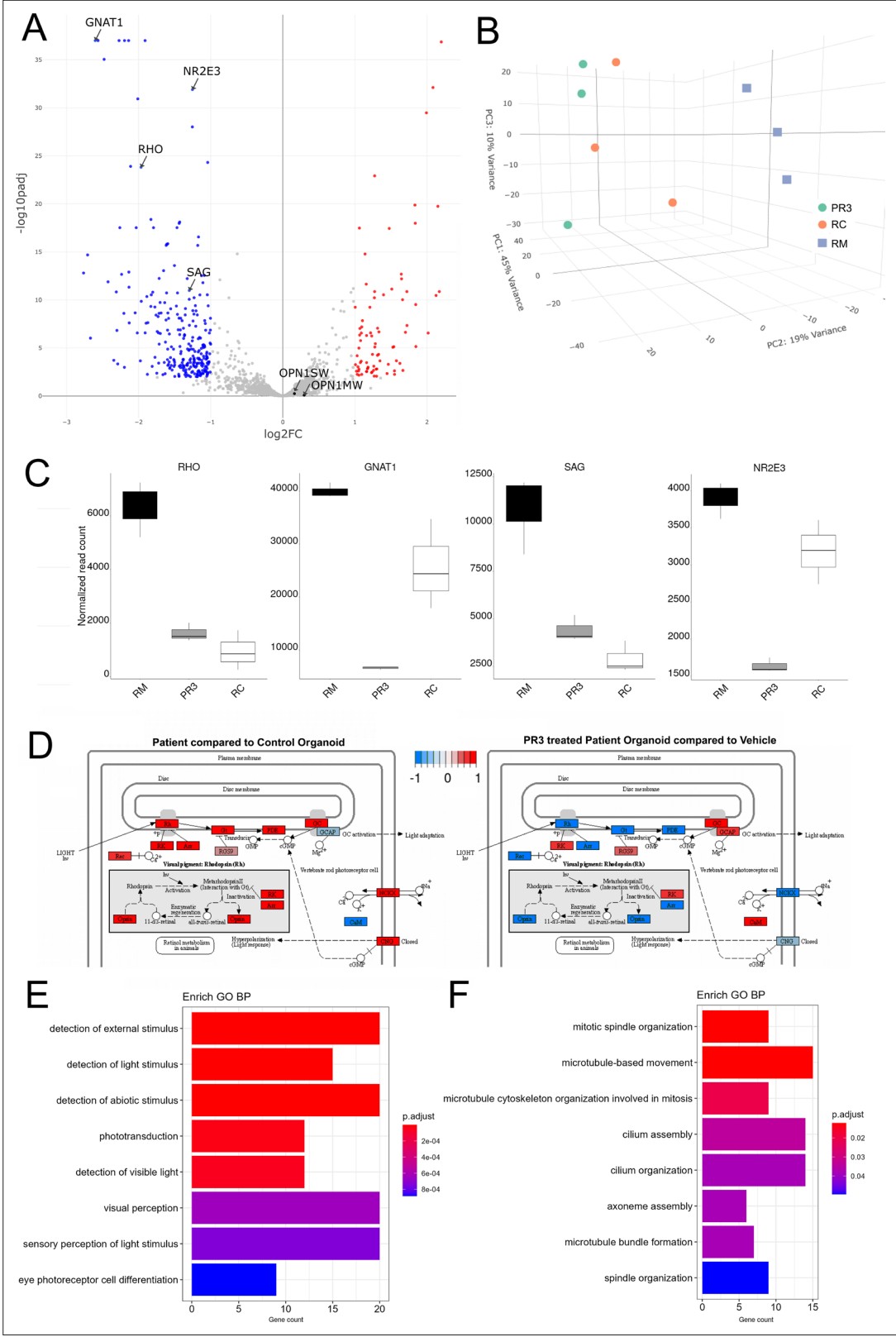

**Figure 6.** RNA sequencing analysis of *RHO*-CNV organoids following Photoregulin3 (PR3) treatment. (**A**) Volcano plot showing significant differentially expressed genes following 1 week of 0.25 μM PR3 treatment in D300+ patient organoids compared to vehicle treated with cutoffs at p<0.01 and ~1Log2FC. (**B**) 3D principal component analysis (PCA) plot showing the tightly clustered independent biological replicates from the control (RC) organoids,

*Figure 6 continued*

vehicle-treated patient (RM) organoids, and PR3-treated patient organoids (PR3). PR3-treated patient organoids were spatially closer to control (RC) compared to patient (RM) organoids. (**C**) Normalized read count plots showing relative expression of *RHO, GNAT1, SAG,* and *NR2E3* in the three conditions. (**D**) KEGG analysis of differentially expressed genes showing dysregulation of key phototransduction pathway comparing RC with RM organoids and the recovery following PR3 treatment in RM organoids. Down- and upregulated genes are indicated in blue and red respectively. (**E, F**) Box plots showing EnrichGO analysis of differentially expressed genes that are either downregulated (**E**) or upregulated (**F**) by comparing PR3 treated to vehicle-treated RM organoids. N=3 independent experiments and 12–15 organoids per experiment. Log2FC = Log2Fold change. CNV = copy number variation.

The online version of this article includes the following figure supplement(s) for figure 6:

**Figure supplement 1.** Recovery of glycosylation and endoplasmic reticulum (ER)-Golgi transport pathways in Photoregulin3 (PR3)-treated *RHO*-CNV organoids in the RNA sequencing analysis.

---

overwhelming the transport machinery, thereby mislocalizing and driving cell dysfunction and death, as observed with Q344R/P/ter *RHO* mutations in mice (**Sung et al., 1994**) (b) reducing the supply of phototransduction deactivating proteins for chromophore-attached RHO, leading to constitutive activation and degeneration, as proposed in some forms of *RHO* mutations such as G90D and T94I (**Dizhoor et al., 2008**) (c) reducing RHO glycosylation in the Golgi leading to RHO instability and photoreceptor degeneration, as in T17M mutation (**Murray et al., 2015**), or by (d) RHO misfolding, driving ER stress/UPR (**Olsson et al., 1992**; **Chiang et al., 2015**). Based on the RNA sequencing data of our current study, it is likely that RHO overexpression overloads the rod ER, by significantly reducing the efficiency of RHO glycosylation. Previous studies have shown that RHO undergoes N-linked glycosylation at two distinct sites, Asn-2 and Asn-15. This glycosylation is believed to be critical for protein folding through interactions with chaperones as well as for the transport of rhodopsin to the outer segments. Interestingly, studies in *Xenopus* suggest that this glycosylation is also responsible for nascent disc assembly (**Murray et al., 2009**). While we had initially hypothesized that RHO-CNV could lead to photoreceptor dysfunction due to ER stress/UPR dysfunction, we did not detect any such changes.

Mislocalization of RHO to the soma was the most striking phenotype discerned in our patient organoids. While there were some breakthroughs of RHO in putative OS, most of the expression especially in mature organoids (>D250) was stuck in the soma. Similar trafficking defects have been documented in post-mortem patient eye samples. RHO protein was found to be localized in the cell soma as well as synaptic processes and neurite extensions in RHO adRP patient samples as well as other RP patients (**Li et al., 1995**). This was also observed in transgenic pigs expressing human mutant RHO (Pro347Leu) (**Li et al., 1998**) as well as *Xenopus* expressing Q350ter (**Tam et al., 2006**). A remarkable feature of adRP is the late-stage retinal disease manifesting slow progressive degeneration with initial OS disruption prior to photoreceptor death. In our results, the *RHO*-CNV patient-matched retinal organoid model rigorously mimicked the clinically degenerative OS and intact photoreceptor phenotype. Interestingly, the mislocalized RHO was not degraded over extended culture time of 300 days. Transgenic Rho mutant mice such as P23H mice have demonstrated the degradation of mislocalized RHO at 3 months of age (**Price et al., 2011**). These findings suggest that the death of the photoreceptors and vision loss in *RHO*-CNV patient is conversely different from patients with other adRP mutations including the P23H mutations. Hence it is essential to understand the underlying mechanism of rod photoreceptor death with similar mislocalized RHO cellular phenotype in patient-specific P23H and *RHO*-CNV retinal organoid models. Proof-of-principle studies utilizing PR3 have been established to alleviate the retinal pathology. Hence further studies comparing the P23H and *RHO*-CNV retinal organoid models could serve as a valuable cellular platform to evaluate the efficacy of potential genome editing and small molecule therapeutic strategies.

There is an increasing interest in gene augmentation therapeutic strategies for retinal degeneration patients with biallelic mutations in *RPE65*, since the approval of Voretigene neparvovec. Although visual outcomes in most treated patients have been very encouraging with significant improvement in visual field and light sensitivity (**Maguire et al., 2021**), there have been some recent reports of progressive pericentral atrophy (**Gange et al., 2022**). One potential cause of RPE atrophy following *RPE65* gene augmentation could be overexpression of *RPE65* in the AAV-*RPE65* infected cells, due

to the utilization of strong promoters such as CAG. Furthermore, the current preclinical approaches to target adRP due to *RHO* are primarily based on gene therapies, with three competing approaches: one to knock down the mutant *RHO* by replacing with a wild-type *RHO*, second to merely overexpress wild-type RHO to overcome the mutant protein effect (*Massengill and Lewin, 2021*), and third to overexpress NR2E3, the upstream regulator of *RHO* (*Liu et al., 2021*). Overexpression of wild-type RHO approach could have unintended adverse consequences on photoreceptor survival if the relationship between RHO expression and rod survival is not clearly understood. Our current study on *RHO*-CNV associated with adRP raises the awareness of toxicities and complications that might arise from either of the gene augmentation therapy approaches.

Nuclear receptors are ideal therapeutic targets because their activities can be readily induced or repressed with small molecules. This allows for fine-tuning of the biological functions of the receptor to alter disease pathogenesis. A few drugs targeting these receptors are already in the clinic (*Dhiman et al., 2018*). It is interesting that targeting a reduction in wild-type RHO expression has also been proposed as a pan-RP therapy. Several groups have explored reduction in the expression of RHO as even small decreases can slow RP progression (*Lewin et al., 1998*). The photoregulin group of compounds, identified through a screen to target the *NR2E3* orphan nuclear receptor, were shown to repress rhodopsin expression with minor effects on cone opsins in mice (*Nakamura et al., 2017*). Photoregulins are suggested to act by enhancing the ability of NR2E3 to complex with NRL and thereby prevent NRL and CRX from interacting with other DNA consensus sites or binding partners though effects on other nuclear orphan receptors were not completely ruled out. Additionally, PR3 was shown to reduce rhodopsin expression in wild-type mice but not in rd7 mice bearing Nr2e3 mutations. Upon delivery to mice by intraperitoneal injection, PR3 has been found to be safe and effective in mitigating retinal degeneration, and improving visual function in the P23H rhodopsin

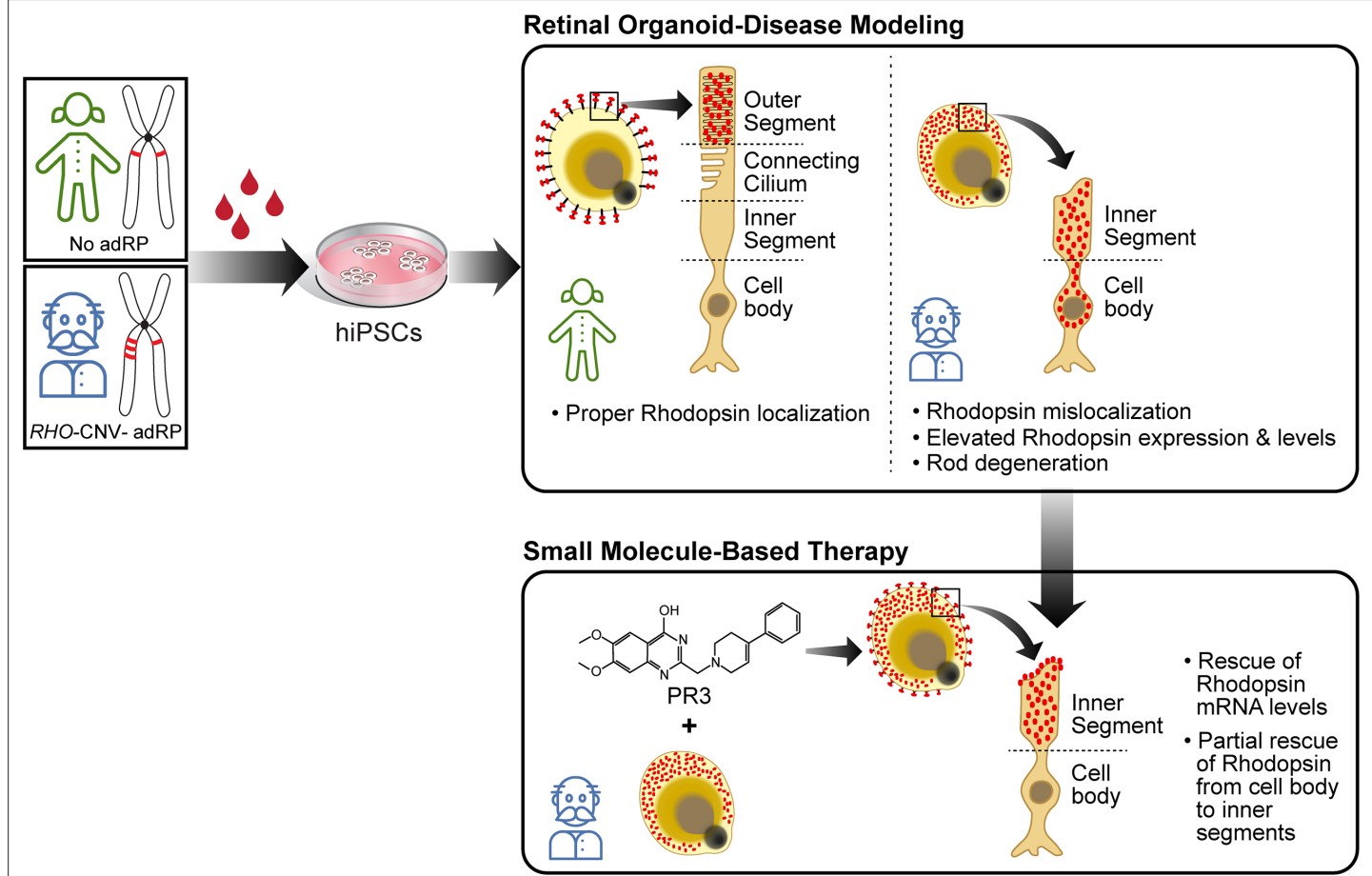

**Figure 7.** Cartoon showing key highlights of our study. We utilized the potential of using patient blood derived induced pluripotent stem cells (iPSCs) for generating retinal organoids for disease modeling studies and screening therapeutics to modulate disease pathogenesis.

mouse model (*Nakamura et al., 2017*), suggesting photoregulins are attractive preclinical candidates to treat humans with *RHO*-related adRP. In our current study, PR3 had a robust effect on rhodopsin expression with no effects on cone opsins and could be a very useful therapeutic for patients with CNV. However, in the case of other adRP *RHO* mutations, careful individualized dose management of the drug will likely be required to achieve adequate reduction of mutant rhodopsin expression in cases of adRP such that it no longer causes a toxic dominant negative effect while still allowing adequate wild-type RHO expression to allow for normal rod function as complete loss of rhodopsin can cause recessive RP.

In conclusion, we have established a disease-in-a-dish model for *RHO*-CNV-associated adRP, and a potential therapeutic option for managing the devastating outcomes of the disorder (*Figure 7*).

## Conclusion

This is the first reported study of characterizing the *RHO*-CNV, a novel cause of adRP. This case of CNV has expanded the profile of highly mutated *RHO* by correlating the clinical behavior to disease modeling using retinal organoids. We have used the organoid model for targeted drug testing with PR3, small molecule inhibitor of *NR2E3*. Taken together, the results of this study demonstrate that *RHO* expression requires precise control for rod photoreceptor functioning and maintenance.

## Methods

### Clinical and molecular diagnosis

The proband (unidentified Lab ID # RM) and his asymptomatic daughter (unidentified Lab ID # RC) were examined at the University of California San Francisco (UCSF, CA, USA) including the pedigree data collection, clinical examination, and genetic diagnosis. Retinal examination included visual acuity testing, fundus photography, SD-OCT, full-field ERG testing. For genetic diagnosis, peripheral blood (PB) samples collected from the proband and the unaffected daughter of the proband were screened by targeted NGS at Blueprint Genetic, a College of American Pathologists – and Clinical Laboratory Improvement Amendments – certified laboratory. The details of the methodology are available in our previous publication (*Tuupanen et al., 2022*). In brief, the sequencing was conducted using a retinal dystrophy NGS panel consisting of 266 genes, which was derived from an in-house tailored whole exome sequencing assay. The sequence reads were mapped to the human reference genome (GRCh37/hg19). CNV analysis was conducted using CNVkit (*Talevich et al., 2016*). Following a manual review of the identified RHO-CNV, discordant read-pairs suggesting a duplication-inverted triplication-duplication event was detected (*Carvalho et al., 2011*).

### Generating iPSC lines

PB samples collected from the human subjects (unidentified Lab ID # RC and RM) were reprogrammed into iPSCs as per the schematic outlined in *Figure 2—figure supplement 1A*. Blood samples were processed by density-gradient centrifugation using Ficoll-paque PLUS (17-1440-02, GE Healthcare Biosciences, Sweden) to isolate the PBMNCs (*Figure 2—figure supplement 1B*). A small fraction of ~1–2 × 10$^6$ PBMNCs was cultured in 1-well of a 12-well suspension plate with PBMNCs expansion medium (*Supplementary file 1a*) by changing half media every day. Date of seeding the PBMNCs in culture was designated as day (D) –4. Four days later, ~0.2–0.4 × 10$^6$ PBMNCs were reprogrammed using CytoTune-iPS 2.0 Sendai Reprogramming Kit (A16517, Thermo Fisher Scientific, USA), at 2.5:2.5:1.5 (KOS:c-myc: Klf4) multiplicity of infection. 24 hr post transfection on D1, a full media change was done. Two days post transfection on D3, cells were plated onto one to two wells of a Matrigel (354234, Corning, USA, diluted as 1 µg/mL in DMEM/F12)-coated six-well plate with reprogramming medium (*Supplementary file 1a*). Half media change was done every other day from D3 until D6. For all the suspension cultures, medium change was done by centrifugation at 200 × *g* for 10 min and resuspending the cell pellet with the respective medium and culturing back into the plates. On D8, cells were transitioned to iPSC medium (*Supplementary file 1a*) with further media changes done on every other day. About 14–21 days post transfection, colonies with typical characteristics of iPSC morphology were manually picked. Pure, compact, and tightly packed iPSC clones were lifted using dissociating solution (*Supplementary file 1a*) for further expansion, characterization

(*Figure 2—figure supplement 1C, D*), and retinal organoid differentiation. Cell Lines available upon request following UCSF MTA guidelines.

## Retinal organoid differentiation

Retinal organoids were differentiated (three to six independent differentiation from each clones) using three different clones from each iPSC lines via the three step embryoid body (EB) approach following our previously published protocols and as per the schematic depicted in *Figure 3A* (*Bachu et al., 2022*; *Arthur et al., 2023*). Briefly, iPSC colonies were lifted and cultured in six-well suspension plate. Small iPSC colonies self-aggregated as EBs within 24 hr were gradually transitioned to complete Neural Induction Medium (NIM) (*Supplementary file 1a*) from D0 to D3. On D6, EBs were treated with 1.5 nM BMP4 (120-05ET, PeproTech, USA). By D7, the EBs were transferred onto a Matrigel-coated plate to facilitate the adherence of EBs to the plate with NIM and 1.5 nM BMP4. Brief exposure of BMP4 in the differentiation process included the treatment of EBs with 1.5 nM (from D6 to D8) to 0.75 nM (D9 to D11) and 0.375 nM (D12 to D14). On D15, BMP4 was completely removed and EBs were fed with just NIM. From D16 through D30, the adherent EBs were fed Retinal Differentiation Medium (RDM) (*Supplementary file 1a*) at the intervals of every 2 days. Regions of the plate displaying clear, shiny borders indicative of retina-like morphology were manually lifted using a sterile P100 pipette tip. Lifted neural retina was transferred to a six-well suspension plates and allowed to self-acquire the 3D organoid configuration within the next 2 days. From D31, the developing retinal organoids were fed 3D-RDM (*Supplementary file 1a*) along with 1 µM all-trans retinoic acid (R2625, Millipore Sigma) every 3 days until D120. From D120 onward, organoids were fed with just 3D-RDM without all-trans retinoic acid every 3 days until the completion of the study. Organoids were periodically assessed for morphological characteristics using phase contrast microscopy (Olympus IX70) at regular intervals. Various time-points of rod photoreceptor differentiation and maturation in the retinal organoids were utilized for phenotypic characterization and drug assessments (*Figure 4—figure supplement 1A-D*).

## Western blotting

Retinal organoids were collected and washed in cold 1× PBS twice at 5 min intervals. Protein was extracted by homogenizing the organoids using hand-held pestle (1415-5390, USA Scientific) in ice-cold RIPA buffer (*Supplementary file 1b*) with 1% cOmplete Protease Inhibitor Cocktail (11697498001, Roche Diagnostics, GmbH). Lysates were incubated at 4°C for 10 min and then centrifuged at 14,000 × $g$ for 15 min at 4°C. Supernatant was collected and the protein concentration was measured using Pierce BCA Protein Assay Kit (23227, Thermo Scientific, USA) as per the manufacturer's instructions. Equal amounts (20 µg) of protein lysates were mixed with 4× Laemmli sample buffer (161-0747, Bio-Rad, USA) and 10% dithiothreitol (DTT, R0861, Thermo Scientific, USA). Samples were resolved on a precast gel (12% Mini-Protean TGX Gels, 456-1044, Bio-Rad, USA) for 20 min at 70 V until the loading dye entered the resolving layer, then increased to 150 V for 40–60 min or until the run was completed (*Supplementary file 1b*). After electrophoresis, the proteins were transferred onto an Immobilon-FL PVDF membrane (IPFL20200, Merck Millipore, USA) ice-cold transfer buffer (*Supplementary file 1b*) at 100 V for 90 min. The blots were blocked with blocking solution for 30 min. The blots were incubated overnight at 4°C with primary antibodies diluted in blocking solution (*Supplementary file 1c* for primary antibody details). Thereafter, blots were washed thrice with TBS-T buffer (10 min interval each) and incubated for 1 hr at room temperature with host-specific secondary antibodies diluted in blocking solution (*Supplementary file 1d* for secondary antibody details). The blots were washed another three times with TBS-T buffer (10 min interval each) and visualized using LiCor Odyssey XF scanner. For quantification of protein expression, the background was subtracted for each band and normalized to internal control band using ImageJ software. Statistical calculations were performed using multiple technical and five independent biological replicates.

## Immunohistochemistry and imaging

Retinal organoids and confluent colonies of iPSCs were fixed in 4% paraformaldehyde (PFA) (157-8, Electron Microscopy Sciences, USA) in 1× PBS for 20 min. Organoids were cryopreserved in 15% through 30% sucrose (made in 1× PBS), and frozen in 2:1 mixture (20% sucrose: OCT, Sakura, USA). 10 µm sections of retinal organoids were collected on Super frost Plus microscopic slides (12-550-15, Fisher Scientific, USA) using Leica CM3050 S cryostat. Immunohistochemistry was done as described

previously. Organoid sections were pap-pen and PFA-fixed iPSC clones were permeabilized at room temperature for 15 min with 0.1% Triton X-100 (0694-1L, VWR Life Science, USA) in 10% Normal Donkey Serum (NDS, S30-100ML, EMD Millipore Corp, USA; made in 1× PBS). Following to that, sections were incubated for 1 hr in 10% NDS. Primary antibodies (*Supplementary file 1c*) diluted in 10% NDS were added to the slides and incubated overnight (12–18 hr) at 4°C. The slides were washed thrice in 1× PBS at 5 min intervals and further incubated for 1 hr at room temperature with fluorescently conjugated secondary antibodies (*Supplementary file 1d*) diluted at 1:250 in 10% NDS. Cell nuclei were counterstained with DAPI (1 μg/mL, Roche, USA) for 10 min. The slides were then washed thrice in 1× PBS at 5 min interval and coverslipped using Fluoromount G (Electron Microscopy Sciences, USA). TUNEL analysis was carried out per the manufacturer's protocol (Sigma-Aldrich). Images were acquired using ZEN software on LSM700 confocal microscope (Zeiss, Inc) and processed by ImageJ (NIH, USA).

## RNA extraction and qRT-PCR

Total RNA was extracted from retinal organoids using RNeasy Mini Kit (74104, QIAGEN, USA) as per the manufacturer's instructions. The quality and purity of the extracted RNA was assessed by NanoDrop One (Thermo Fisher Scientific, USA). cDNA was synthesized using iScript cDNA Synthesis Kit (1708891, Bio-Rad, USA). qRT-PCR was performed on CFX96 system (Bio-Rad, USA) using iTaq Universal SYBR Green Supermix (64047467, Bio-Rad, USA). Primer sequences used for qRT-PCR are listed in *Supplementary file 1e*. The amplification reaction sets were: 95°C for 30 s, 40 cycles at 95°C for 5 s, 60°C for 25 s, 95°C for 5 s, and final extension at 95°C for 5 s. The Ct values of the target genes were first normalized to the endogenous control β-actin. The corrected Ct values were then utilized to compare and validate the control and patient retinal organoids. $Log_2$ fold change (Log2FC) in gene expression of all targets were then calculated.

## Bulk RNA sequencing

The quality control, RNA library preparations, and sequencing reactions of the extracted RNA from retinal organoids were performed at Novogene Corporation Inc (CA, USA). FASTQ files received from Novogene following QC were quantified using Salmon package (v1.9.0) by pseudo-aligning against *Homo sapiens* hg19 genomic assembly in Galaxy (*Afgan et al., 2016*). Low counts (<10) were filtered out and batch correction was carried out using Combat package in DEBrowser (v1.24.1) R package (*Kucukural et al., 2019*). Differential expression analysis was carried out using DESeq2 in DEbrowser. Genes with an adjusted p-value <0.05 were assigned as differentially expressed. Gene ontology enrichment analysis and KEGG pathway analysis of differentially expressed genes were carried out in DEBrowser and Beavr (*Perampalam and Dick, 2020*) packages. Differential expression analysis and pathway enrichment data have been uploaded as supplementary files (*Supplementary file 2*). The data is uploaded to NCBI GEO database (GSE245545).

## Transmission electron microscopy

Retinal organoids (300 days of age) were washed in 1× PBS three times at 10 min intervals and fixed with Karnovsky's fixative (4% PFA/2% glutaraldehyde in 0.1 M PBS, pH 7.4) overnight at 4°C. For TEM, organoids were washed in 1× PBS and stained in 1% osmium tetroxide (19180, Electron Microscopy Sciences, USA; made in distilled water) for 1 hr at room temperature followed by staining with 2% uranyl acetate (22400, Electron Microscopy Sciences, USA; made in distilled water). After three washes in distilled water, organoids were then dehydrated in a graded series of ice-cold ethanol (50%, 70%, 95%, and 100%) for 20 min each. Organoids were incubated in propylene oxide (00235-1, Polysciences Inc, USA) twice for 5 min, followed by a 5 min incubation in 1:1 mix of propylene oxide: Epon 812 (13940, Electron Microscopy Sciences, USA). Organoids were allowed to infiltrate in Epon 812 overnight at room temperature. Next day, the organoids were embedded in PELCO silicone rubber molds (105, Ted Pella Inc, USA) using Epon 812, and polymerized for 48 hr at 60°C. Ultrathin (70 nm) sections were collected and imaged using a Philips Tecnai 10 electron microscope at the VA Medical Center, San Francisco, USA.

## PR3 treatment on retinal organoids

Patient-specific *RHO*-CNV retinal organoids (~300 days of age) were treated with small drug-like molecule PR3 (SML2299-5MG, Sigma, 5 mM stock made in DMSO) supplemented in 3D-RDM for

1 week by changing medium every other day. Three different concentrations of PR3 utilized in this study were 0.1, 0.25, and 0.5 µM. DMSO was used as a vehicle control. One week, post PR3 treatment, retinal organoids were collected and assessed by qRT-PCR, bulk RNA sequencing, and imaging experiments.

## Statistical analysis

All the data were obtained from three to six independent retinal differentiation experiments. The quantified data values were provided as mean ± SEM. The intergroup differences for all the analysis were determined with the GraphPad Prism v9, using a two-tailed Student's t-test or ANOVA. Significant differences were indicated by p-values listed in the figures. For imaging studies, at least three sections were averaged to account for regional variability in iPSC reprograming and retinal organoid differentiation.

## Acknowledgements

We are grateful to the recruited human subjects and families included in this study. We thank members of the Lamba lab for helpful discussions and suggestions. The research presented here is supported by the R01 EY032197 and CIRM DISC0-14449 to DAL, Foundation Fighting Blindness research grant to DAL, U24 EY029891 to DAL and JLD, Postdoctoral fellowship from All May See Foundation and Bright Focus Foundation macular degeneration research to SK, UCSF Vision Core NIH/NEI P30 EY002162, and unrestricted grant from Research to Prevent Blindness, New York, NY, to Department of Ophthalmology at UCSF. We deeply appreciate the help of Yien-Ming Kuo at the UCSF Vision Core and Ivy Hsieh at VA Medical Center for processing the electron microscopy samples. We would also like to thank Suling Wang for drawing all the illustrations included in this paper based on the descriptions given by the authors. Additionally, we would like to acknowledge the contributions of The Foundation Fighting Blindness My Retina Tracker Genetic Testing Program for identifying the *RHO*-CNV in the patient. We deeply appreciate the services given by Blueprint Genetics who undertook the genetic analysis and particularly Sari Tuupanen who resolved the complex genetic rearrangement within the *RHO* locus.

## Additional information

### Competing interests

Jacque L Duncan: Dr. Duncan was a consultant for ConeSight, DTx Therapeutics, Editas,Eloxx, Eyevensys, Gyroscope, Helios,Nacuity, ProQR, PYC Therapeutics,Replay Therapeutics, Spark,SparingVision, Vedere Bio until 01/2022. These were unrelated to the manuscript. Deepak A Lamba: is affiliated with Genentech. The author has no financial interests to declare. The other authors declare that no competing interests exist.

### Funding

| Funder | Grant reference number | Author |
|---|---|---|
| California Institute for Regenerative Medicine | DISC0-14449 | Deepak A Lamba |
| National Eye Institute | EY032197 | Deepak A Lamba |
| Foundation Fighting Blindness | | Deepak A Lamba |
| National Eye Institute | EY029891 | Jacque L Duncan Deepak A Lamba |
| All May See Foundation | | Sangeetha Kandoi L Vinod K Reddy |
| BrightFocus Foundation | | Sangeetha Kandoi |
| National Eye Institute | EY002162 | Jacque L Duncan |

| Funder | Grant reference number | Author |
|---|---|---|
| Research to Prevent Blindness | | Jacque L Duncan |

The funders had no role in study design, data collection and interpretation, or the decision to submit the work for publication.

## Author contributions

Sangeetha Kandoi, Conceptualization, Data curation, Formal analysis, Validation, Investigation, Visualization, Methodology, Writing – original draft, Writing – review and editing; Cassandra Martinez, Formal analysis, Validation, Investigation, Visualization; Kevin Xu Chen, Miika Mehine, Data curation, Formal analysis, Investigation, Methodology; L Vinod K Reddy, Data curation, Formal analysis, Investigation, Visualization; Brian C Mansfield, Conceptualization, Writing – review and editing; Jacque L Duncan, Conceptualization, Formal analysis, Methodology, Project administration, Writing – review and editing; Deepak A Lamba, Conceptualization, Resources, Data curation, Formal analysis, Supervision, Funding acquisition, Investigation, Visualization, Methodology, Writing – original draft, Project administration, Writing – review and editing

## Author ORCIDs

Jacque L Duncan ⓘ https://orcid.org/0000-0002-9593-6412
Deepak A Lamba ⓘ https://orcid.org/0000-0002-2811-307X

## Ethics

The participants in the study gave written consent to participate in the current study and have the results of this research work published. All written informed consent was approved by the UCSF Institutional Review Board (IRB # 18-26409) and adhered to the tenets set forth in the Declaration of Helsinki.

Reviewer #2 (Public Review): https://doi.org/10.7554/eLife.90575.3.sa1
Reviewer #3 (Public Review): https://doi.org/10.7554/eLife.90575.3.sa2
Author response https://doi.org/10.7554/eLife.90575.3.sa3

# Additional files

## Supplementary files

• Supplementary file 1. Formulations of media (**1a**) and buffers (**1b**). Details of primary (**1c**) and secondary (**1d**) antibodies used in the manuscript. Primers used for the RT-PCR analysis (**1e**).

• Supplementary file 2. Excel file of differentially expressed genes from bulk RNA sequencing analysis.

• MDAR checklist

## Data availability

All data generated and analyzed are included in the manuscript, figures and figure supplements. Sequencing data is uploaded to NCBI GEO database under accession code GSE245545.

The following dataset was generated:

| Author(s) | Year | Dataset title | Dataset URL | Database and Identifier |
|---|---|---|---|---|
| Kandoi S, Martinez C, Chen KX, Reddy LVK, Mehine M, Mansfield BC, Duncan JL, Lamba DA | 2023 | Modeling and modulation of RHO signaling in human iPSC-derived retinal organoids from patients with RHO-CNV | https://www.ncbi.nlm.nih.gov/geo/query/acc.cgi?acc=GSE245545 | NCBI Gene Expression Omnibus, GSE245545 |

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
