## [Editor Report · eLife assessment]

This study presents an **important** finding that implicates a rhodopsin gene duplication in the progression of autosomal dominant retinitis pigmentosa in patients. The authors utilize a retinal organoid model to demonstrate a similar disease phenotype and suggest defects can be ameliorated by using photoregulin. The data supporting the conclusions are **solid**, but there are some concerns regarding the strength of the phenotype in retinal organoids. This work will be of broad interest to vision researchers.

---

## [Referee Report · Reviewer #2 (Public Review)]

Summary:

The manuscript by Kandoi et al. describes a new 3D retinal organoid model of a mono-allelic copy number variant of the rhodopsin gene that in a patient led to autosomal dominant retinitis pigmentosa. The evidence provided here is relatively strong that the rod photoreceptor phenotype observed in an adult patient with RP in vivo is similar to that phenotype observed in human stem cell-derived retinal organoids. Increases in RHO expression were detected by qPCR, RNA-seq, and IHC support this phenotype. Importantly, the amelioration of photoreceptor rhodopsin mislocalization and related defects using the small molecule drug photoregulin demonstrates an important potential clinical application.

Strengths:

- Retinal organoids derived from patient with adRP.

- RHO mislocalization could explain the phenotype in patients.

Weaknesses:

- Organoids at 300 days do not show PR loss.

Additional minor weaknesses

- Bulk RNAseq methods require greater detail, particularly with respect to how total or mRNA was purified, how was it quantified for concentration and integrity (i.e. Nanodrop, Tape station, Bioanalyzer), what reagents were used for library preparation and how many reads were analyzed per sample.

- Fig. 4. The levels of RHO visualized in tissue sections (panels A-C) does not seem to match the general levels shown for the western blots (panel D) which appear to be far higher in RM western blot samples than in the IHC images. Please clarify why there is such a difference.

- Line 186: by what criteria are the authors able to state that " there were no clear visible anatomical changes in apical-basal retinal cell type distribution (data not shown)". Was this based on histological staining with antibodies, nuclear counter-staining or some other evaluation?

---

## [Referee Report · Reviewer #3 (Public Review)]

This manuscript reports a novel pedigree with four intact copies of RHO on a single chromosome which appears to lead to overexpression of rhodopsin and a corresponding autosomal dominant form of RP. The authors generate retinal organoids from patient- and control-derived cells, characterize the phenotypes of the organoids, and then attempt to 'treat' aberrant rhodopsin expression/mislocalization in the patient organoids using a small molecule called photoregulin 3 (PR3). While this novel genetic mechanism for adRP is interesting, the organoid work is not compelling. There are multiple problems related to the technical approaches, the presentation of the results, and the interpretations of the data. I will present my concerns roughly in the order in which they appear in the manuscript and will separate them into 'major' and 'minor' categories:

Major concerns:

(1) Individual human retinal organoids in culture can show a wide range of differentiation phenotypes with respect to the expression of specific markers, percentages of given cell types, etc. For this reason, it can be very difficult to make rigorous, quantitative comparisons between 'wild-type' and 'mutant' organoids. Despite this difficulty, the author of the present manuscript frequently present results in an impressionistic manner without quantitation. Furthermore, there is no indication that the investigator who performed the phenotypic analyses was blind with respect to the genotype. In my opinion, such blinding is essential for the analysis of phenotypes in retinal organoids.

To give an example, in lines 193-194 the authors write "we observed that while the patient organoids developing connecting cilium and the inner segments similar to control organoids, they failed to extend outer segments". Outer segments almost never form normally in human retinal organoids, even when derived from 'wild-type' cells. Thus, I consider it wholly inadequate to simply state that outer segment formation 'failed' without a rigorous, quantitative, and blinded comparison of patient and control organoids.

(2) The presentation of qPCR results in Fig. 3A in very confusing. First, the authors normalize expression to that of CRX, but they don't really explain why. In lines 210-211 they write "CRX, a ubiquitously expressing photoreceptor gene maintained from development to adulthood." Several parts of this sentence are misleading or incomplete. First, CRX is not 'ubiquitously expressed' (which usually means 'in all cell types') nor is it photoreceptor-specific: CRX is expressed in rods, cones, and bipolar cells. Furthermore, CRX expression levels are not constant in photoreceptors throughout development/adulthood. So, for these reasons alone, CRX is a poor choice for normalization of photoreceptor gene expression.

Second, the authors' interpretation of the qPCR results (lines 216-218) is very confusing. The authors appear to be saying that there is a statistically significant increase in RHO levels between D120 and D300. However, the same change is observed in both control and patient organoids and is not unexpected, since the organoids are more mature at D300. The key comparison is between control and patient organoids at D300. At this time point, there appears to be no difference control and patient. The authors don't even point this out in the main text.

Third, the variability in number of photoreceptor cells in individual organoids makes a whole-organoid comparison by qPCR fraught with difficulty. It seems to me that what is needed here is a comparison of RHO transcript levels in isolated rod photoreceptors.

(3) I cannot understand what the authors are comparing in the bulk RNA-seq analysis presented in the paragraph starting with line 222 and in the paragraph starting with line 306. They write "we performed bulk-RNA sequencing on 300-days-old retinal organoids (n=3 independent biological replicates). Patient retinal organoids demonstrated upregulated transcriptomic levels of RHO... comparable to the qRT-PCR data." From the wording, it suggests that they are comparing bulk RNA-seq of patient and control organoids at D300. However, this is not stated anywhere in the main text, the figure legend, or the Methods. Yet, the subsequent line "comparable to the qRT-PCR data" makes no sense, because the qPCR comparison was between patient samples at two different time points, D120 and D300, not between patient and control. Thus, the reader is left with no clear idea of what is even being compared by RNA-seq analysis.

Remarkably, the exact same lack of clarity as to what is being compared plagues the second RNA-seq analysis presented in the paragraph starting with line 306. Here the authors write "We further carried out bulk RNA-sequencing analysis to comprehensively characterize three different groups of organoids, 0.25 μM PR3-treated and vehicle-treated patient organoids and control (RC) organoids from three independent differentiation experiments. Consistent with the qRT-PCR gene expression analysis, the results showed a significant downregulation in RHO and other rod phototransduction genes." Here, the authors make it clear that they have performed RNA-seq on three types of sample: PR3-treated patient organoids, vehicle-treated patient organoids, and control organoids (presumably not treated). Yet, in the next sentence they state "the results showed a significant downregulation in RHO", but they don't state what two of the three conditions are being compared! Although I can assume that the comparison presented in Fig. 6A is between patient vehicle-treated and PR3-treated organoids, this is nowhere explicitly stated in the manuscript.

(4) There are multiple flaws in the analysis and interpretation of the PR3 treatment results. The authors wrote (lines 289-2945) "We treated long-term cultured 300-days-old, RHO-CNV patient retinal organoids with varying concentrations of PR3 (0.1, 0.25 and 0.5 μM) for one week and assessed the effects on RHO mRNA expression and protein localization. Immunofluorescence staining of PR3-treated organoids displayed a partial rescue of RHO localization with optimal trafficking observed in the 0.25 μM PR3-treated organoids (Figure 5B). None of the organoids showed any evidence of toxicity post-treatment."

There are multiple problems. First, the results are impressionistic and not quantitative. Second, it's not clear that the investigator was blinded with respect to treatment condition. Third, in the sections presented, the organoids look much more disorganized in the PR3-treated conditions than in the control. In particular, the ONL looks much more poorly formed. Overall, I'd say the organoids looked considerably worse in the 0.25 and 0.5 microM conditions than in the control, but I don't know whether or not the images are representative. Without rigorously quantitative and blinded analysis, it is impossible to draw solid conclusions here. Lastly, the authors state that "none of the organoids showed any evidence of toxicity post-treatment," but do not explain what criteria were used to determine that there was no toxicity.

(5) qPCR-based quantitation of rod gene expression changes in response to PR3 treatment is not well-designed. In lines 294-297 the authors wrote "PR3 drove a significant downregulation of RHO in a dose-dependent manner. Following qRT-PCR analysis, we observed a 2-to-5 log2FC decrease in RHO expression, along with smaller decreases in other rod-specific genes including NR2E3, GNAT1 and PDE6B." I assume these analyses were performed on cDNA derived from whole organoids. There are two problems with this analysis/interpretation. First, a decrease in rod gene expression can be caused by a decrease in the number of rods in the treated organoids (e.g., by cell death) or by a decrease in the expression of rod genes within individual rods. The authors do not distinguish between these two possibilities. Second, as stated above, the percentage of cells that are rods in a given organoid can vary from organoid to organoid. So, to determine whether there is downregulation of rod gene expression, one should ideally perform the qPCR analysis on purified rods.

(6) In Fig. 4B 'RM' panels, the authors show RHO staining around the somata of 'rods' but the inset images suggest that several of these cells lack both NRL and OTX2 staining in their nuclei. All rods should be positive for NRL. Conversely, the same image shows a layer of cells sclerad to the cells with putative RHO somal staining which do not show somal staining, and yet they do appear to be positive for NRL and OTX2. What is going on here? The authors need to provide interpretations for these findings.

Minor concerns:

(1) The writing is poor in many places. Problems include: poor word choice (e.g., 'semi-occasional' is used three times where 'occasional' or 'infrequent' would be better); superfluous use of the definite article in many places (e.g., lines 189-190 "by the light microscopy" should be "by light microscopy"); awkward sentence structures (e.g., lines 208-209: "To equilibrate the data to equivalent the number of photoreceptors in organoids"), opaque expressions (e.g., line 217 "there was a significant ~3 log2 fold change (log2FC)"; why not just say "an ~8-fold change"?); poor proof-reading (Abstract says that 40% of adRP cases are due to mutation in RHO, then the Introduction says the figure is 25%) etc.

(2) The figures are not numbered, which makes it painful for the reviewer to correlate main text call-outs, figure legends, and actual figures. I had to repeatedly count down the list of figures to determine which figure I should be looking at.

(3) In the abstract, the authors suggest that the patient's disease "develops from a dominant negative gain of function" mechanism. I don't agree with this interpretation. Typically 'dominant-negative' refers to an aberrant protein which directly interferes with the function of the normal protein, for example by forming non-functional heterodimers. In the present patient, the disease can be explained by a simple overexpression mechanism, as it has been previously demonstrated in mice that even minimal overexpression of rhodopsin (e.g., ~25% more than normal levels) can led to progressive rod degeneration: PMID: 11222515.

(4) In line 85 the word 'Morphologically' is superfluous and can be deleted.

(5) In the Introduction the authors should more clearly articulate the rationale for using PR3 to treat this patient: because it leads to downregulation of multiple rod genes including RHO. This isn't clearly explained until the Discussion.

(6) The authors mention in several places that PR3 may act via inhibition of NR2E3. Although this was the conclusion of the original publication, the evidence that PR3 acts via Nr2e3 in mice is not solid. The original study (PMID: 29148976) showed that the main effect of PR3 application on mouse retinas is downregulation of numerous rod genes. However, knockout of Nr2e3 in mouse has been shown to have very little effect on rod gene expression, and Nr2e3 mutant rods have largely preserved rod function as demonstrated by scotopic ERGs PMIDs: 15634773, 16110338, 15689355. The primary gene expression defect in Nr2e3 mutant mouse rods is upregulation of a subset of cone genes, a change not observed upon application of PR3 to mouse retinas. For these reasons, I am skeptical that PR3 acts via inhibition of Nr2e3 activity, and I would suggest that the present authors qualify that interpretation.

(7) This mechanistic speculation presented in lines 274-278 is not warranted. Ectopic localization of opsin to the cytoplasmic membrane occurs in a wide range of genetic forms of rod degeneration.

---

## [Author Response]

The following is the authors’ response to the original reviews.

**Public Reviews:**

**Reviewer #1 (Public Review):**
Summary:In the manuscript titled "Disease modeling and pharmacological rescue of autosomal dominant Retinitis Pigmentosa associated with RHO copy number variation" the authors describe the use of patient iPSC-derived retinal organoids to evaluate the pathobiology of a RHO-CNV in a family with dominant retinitis pigmentosa (RP). They find significantly increased expression of rhodopsin, especially within the photoreceptor cell body, and defects in photoreceptor cell outer segment formation/maturation. In addition, they demonstrate how an inhibitor of NR2E3 (a rod transcription factor required for inducing rhodopsin expression), can be used to rescue the disease phenotype.Strengths:The manuscript is very well written, the illustrations and data presented are compelling, and the authors' interpretation/discussion of their findings is logical.Weaknesses:A weakness, which the authors have addressed in the discussion section, is the lack of an isogenic control, which would allow for direct analysis of the RHO-CNV in the absence of the other genetic sequence contained within the duplicated region. As the authors suggest, CRISPR correction of a large CNV in the absence of inducing unwanted on-target editing events in patient iPSCs is often very challenging. Given that they have used a no-disease iPSC line obtained from a family member, controlled for organoid differentiation kinetics/maturation state, and that no other complete disease-causing gene is contained within the duplicated region, it is unlikely that the addition of an isogenic control would yield significantly different results.Aims and conclusions:This reviewer is of the opinion that the authors have achieved their aims and that their results support their conclusions.Discussion:The authors have provided adequate discussion on the utility of the methods and data as well as the impact of their work on the field.

We thank the reviewer for their insightful, and encouraging review of our work that has taken several years to get to current stage.

**Reviewer #2 (Public Review):**
Summary:The manuscript by Kandoi et al. describes a new 3D retinal organoid model of a mono-allelic copy number variant of the rhodopsin gene that was previously shown to induce autosomal dominant retinitis pigmentosa via a dominant negative mechanism in patients. With advancements in the low-cost genomics application to detect copy number variations, this is a timely article that highlights a potential disease mechanism that goes beyond the retina field. The evidence is relatively strong that the rod photoreceptor phenotype observed in an adult patient with RP in vivo is similar to that phenotype observed in human stem cell-derived retinal organoids. Increases in RHO expression detected by qPCR, RNA-seq, and IHC support this phenotype. Importantly, the amelioration of photoreceptor rhodopsin mislocalization and related defects using the small molecule drug photoregulin demonstrates an important potential clinical application.Overall, the authors succeeded in providing solid evidence that copy number variation via a genomic RHO duplication leads to abnormalities in rod photoreceptors that can be partially blocked by photoregulin. However, there are several points that should be addressed that will enhance this paper.Strengths:The use of patient-derived organoids from patients that have visual defects is a major strength of this work and adds relevance to the disease phenotype.The rod phenotype assessed by qPCR, RNA-seq, and IHC supports a phenotype that shares similarities with the patient.The use of a small molecule drug that selectively targets rod photoreceptors, as opposed to cones, is a noteworthy strength.

We thank the reviewers for highlighting the key strengths of the paper.

Weaknesses:(1) The chromosomal segment that was duplicated had 3 copies of RHO in addition to three copies of each of the flanking genes (IFT122, HIF100, PLXND1). Discussion of the involvement of these genes would be helpful. Would duplication of any of these genes alone cause or contribute to adRP? As an example, a missense mutation in IFT122 was previously implicated in photoreceptor loss (PMID: 33606121 PMCID: PMC8519925).

Thank you for your comment. It is an interesting question on the contribution of the other duplicated genes. Of these, IFT122 is particularly interesting as pointed out. We did a thorough survey through literature and our genetic testing partner’s database, BluePrint Genetics. We did not find any human retinal degeneration cases with variants in IFT122. IFT122 has been shown to cause recessive phenotype in dogs and in complete knockout zebrafish model but dominant or overexpression has not been shown to have a phenotype. Interestingly, recessive biallelic IFT122 mutation can cause Cranioectodermal Dysplasia (Sensenbrenner syndrome, PMID: 24689072) and none of these patient exhibited retinal dystrophy. HIF100 is an epigenetic modifier gene while PLXND1 is expressed in endothelial cells. We will include a discussion on this in the revised manuscript.

(2) Related to #1, have the authors considered inserting extra copies of RHO (and/or the flanking genes) of these at a genomic safe harbor site? Although not required, this would allow one to study cells with isogenic-matched genetic backgrounds and would partially address the technical challenge of repairing a 188kb duplication, which as the authors note would be difficult to do. Demonstrating that excess copy numbers in different genetic backgrounds would be a huge contribution to the field. At a minimum, a discussion of the role of the nearby genes should be included.

Thank you for your suggestion. We plan to test the relative role of 1-3 extra copies of RHO driven off a NRL promoter in order to drive it only in rods in our future mechanistic analysis studies. We will include a discussion on the potential role of the other genes in the revised manuscript.

(3) In the patient, the central foveal region was spared suggesting that cones were normal. Was there a similar assessment that cones are unaffected in retinal organoids?

We will include this data in our revised manuscript but overall did not see a cone defect in RHO CNV organoids. Additionally, although it is true that the central foveal region was relatively spared in this patient, the cones are definitely not normal. The macular cones that remain have been damaged by chronic edema, and photoreceptor and RPE atrophy has progressed into the macula, sparing only the foveal cones.

(4) Pathway analysis indicated that glycosylation was perturbed and this was proposed as an explanation as to why rhodopsin was mislocalized. Have the authors verified that there is an actual decrease in glycosylation?

These studies are ongoing. We are currently looking into the details of cellular pathophysiology focusing on RHO trafficking in RHO-CNV including role of glycosylation and other post-translational modifications defects.

(5) Line 182: by what criteria are the authors able to state that " there were no clear visible anatomical changes in apical-basal retinal cell type distribution during the early differentiation timeframe (data not shown)." Was this based on histological staining with antibodies, nuclear counter-staining, or some other evaluation?

This was based on both IHC for various cell type markers and nuclear (DAPI) staining.

(6) Figure 2C - the appearance of the inner segments in RC and RM looks very different from one another. Have the authors ruled out the possibility that the RC organoid cell isn't a cone? In addition, the RM structure has what appears to be a well-defined OLM which would suggest well-formed Muller glia. Do these structures also exist in RC organoids? Typically the OLM does form in older organoids. In addition, was this representative in numerous EM preparations?

For clarification on EM data, we will include additional images in the revision as supplementary data. We have not carefully compared OLM between the patient and control organoids but do observe them in both conditions in the older organoids. The EM preparations were made from multiple organoids from two different batches with consistent results.

(7) What criteria were used to assess cell loss? Has any TUNEL labeling been performed to confirm cell loss? From the existing data, it seems that rod outer segments appear to be affected in organoids. However, it's not clear if the photoreceptors themselves actually die in this model.

TUNEL was used to assess cell loss and it was not significantly different between the control and patient organoids at the timepoints examined. We did not expect a change as the disease in the patient developed over decades.

(8) Figure 5B. The RHO staining in the vehicle-treated sample is perturbed relative to the PR3 treatments as indicated in the text. In the vehicle-treated sample, the number of DAPI-positive cells that are completely negative proximal to the inner segments suggests that there might be non-rod cells there. Have the authors confirmed whether these are cones? Labels would be helpful in the left vehicle panel as the morphology looks very different than the treated samples.

Thank you very much for the various suggestions and these will be included in the revised manuscript version. A number of the cells in the negative regions are OTX2+/NRL- and likely to be cones (Figure 4 A and B). Unfortunately, we do not have a very good cone nuclear marker as RXRγ does not consistently stain mature cones.

(9) It is interesting that in addition to increases in RHO, and photo-transduction, there are also increases in PTPRT which is related to synaptic adhesion. Is there evidence of ectopic neurites that result from PTPRT over-expression?

You are absolutely correct that PTPRT data is very interesting. PTPRT requires similar PTMs like RHO in photoreceptors for its synaptic localization. We did not specifically look at ectopic neurites and test that in the revision. It will interesting to follow-up on its expression pattern to see if it gets processed or localized normally if we can find a working antibody. It is also possible that the gene-expression increase due to feedback upregulation secondary to improper protein processing.

**Reviewer #3 (Public Review):**
This manuscript reports a novel pedigree with four intact copies of RHO on a single chromosome which appears to lead to overexpression of rhodopsin and a corresponding autosomal dominant form of RP. The authors generate retinal organoids from patient- and control-derived cells, characterize the phenotypes of the organoids, and then attempt to 'treat' aberrant rhodopsin expression/mislocalization in the patient organoids using a small molecule called photoregulin 3 (PR3). While this novel genetic mechanism for adRP is interesting, the organoid work is not compelling. There are multiple problems related to the technical approaches, the presentation of the results, and the interpretations of the data. I will present my concerns roughly in the order in which they appear in the manuscript.Major concerns:(1) Individual human retinal organoids in culture can show a wide range of differentiation phenotypes with respect to the expression of specific markers, percentages of given cell types, etc. For this reason, it can be very difficult to make rigorous, quantitative comparisons between 'wild-type' and 'mutant' organoids. Despite this difficulty, the author of the present manuscript frequently presents results in an impressionistic manner without quantitation. Furthermore, there is no indication that the investigator who performed the phenotypic analyses was blind with respect to the genotype. In my opinion, such blinding is essential for the analysis of phenotypes in retinal organoids. To give an example, in lines 193-194 the authors write "we observed that while the patient organoids developing connecting cilium and the inner segments similar to control organoids, they failed to extend outer segments". Outer segments almost never form normally in human retinal organoids, even when derived from 'wild-type' cells. Thus, I consider it wholly inadequate to simply state that outer segment formation 'failed' without a rigorous, quantitative, and blinded comparison of patient and control organoids.

We agree it is challenging to generate outer segments in retinal organoids but we are not the first to show this. This has been demonstrated by multiple independent labs (Mayerl et al PMID: 36206764), Wahlin et al (PMID: 28396597), West at al (PMID: 35334217) including ours (Chirco et al PMID: 34653402). To clarify, we did not observe any OS like tissue in the patient organoids across multiple EM preps of a number of organoids from two independent 300+ day experiments which matched the phase microscopy data presented in Fig2B.

(2) The presentation of qPCR results in Figure 3A is very confusing. First, the authors normalize expression to that of CRX, but they don't really explain why. In lines 210-211, they write "CRX, a ubiquitously expressing photoreceptor gene maintained from development to adulthood." Several parts of this sentence are misleading or incomplete. First, CRX is not 'ubiquitously expressed' (which usually means 'in all cell types') nor is it photoreceptor-specific: CRX is expressed in rods, cones, and bipolar cells. Furthermore, CRX expression levels are not constant in photoreceptors throughout development/adulthood. So, for these reasons alone, CRX is a poor choice for the normalization of photoreceptor gene expression.

As you are aware, all housekeeping genes have shortcomings when used for normalizing PCR data. We went with CRX as within the timepoints chosen, it is not expected to change much and thus represent a good equalizer for relative photoreceptor numbers between the organoids and conditions. While we agree that CRX is weakly expressed in bipolar cells (Yamamoto et al 2020), it is not expected to bias the data too much as we have not seen nor have other reported a huge relative difference in bipolar cell number in organoids. We also confirm this by showing equivalent expression of OTX2, RCVRN and NRL between all conditions.

Second, the authors' interpretation of the qPCR results (lines 216-218) is very confusing. The authors appear to be saying that there is a statistically significant increase in RHO levels between D120 and D300. However, the same change is observed in both control and patient organoids and is not unexpected, since the organoids are more mature at D300. The key comparison is between control and patient organoids at D300. At this time point, there appears to be no difference between control and patient. The authors don't even point this out in the main text.

Thank you for the comment and we apologize if this confused you. However, as can been seen in the graph in Figure 3A, we do compare expression of genes including RHO between control and patient organoids at two different time points. There are four conditions: D120-RC, D120-RM, D300-RC and D300-RM with individual data points and error bars for each condition. There is a statistically significant increase at both time points upon comparing the control and patient organoids for RHO. We compared RHO expression between patient organoids at the two time points and it was not statistically different.

Third, the variability in the number of photoreceptor cells in individual organoids makes a whole-organoid comparison by qPCR fraught with difficulty. It seems to me that what is needed here is a comparison of RHO transcript levels in isolated rod photoreceptors.

We agree that this makes it challenging. This was the exact reasoning for using CRX for normalization since it is predominantly present in photoreceptors. This was validated by the data showing no difference in expression of photoreceptor markers OTX2, RCVRN or NRL between the organoids.

(3) I cannot understand what the authors are comparing in the bulk RNA-seq analysis presented in the paragraph starting with line 222 and in the paragraph starting with line 306. They write "we performed bulk-RNA sequencing on 300-days-old retinal organoids (n=3 independent biological replicates). Patient retinal organoids demonstrated upregulated transcriptomic levels of RHO... comparable to the qRT-PCR data." From the wording, it suggests that they are comparing bulk RNA-seq of patients and control organoids at D300. However, this is not stated anywhere in the main text, the figure legend, or the Methods. Yet, the subsequent line "comparable to the qRT-PCR data" makes no sense, because the qPCR comparison was between patient samples at two different time points, D120 and D300, not between patient and control. Thus, the reader is left with no clear idea of what is even being compared by RNA-seq analysis.

We apologize if the conditions were not obvious and will clarify this in the revised version. The conditions compared are control and patient organoids at D300. Regarding comparison to RT-PCR, as stated above, the comparison shown is between patient and control organoids at two different timepoints.

Remarkably, the exact same lack of clarity as to what is being compared is found in the second RNA-seq analysis presented in the paragraph starting with line 306. Here the authors write "We further carried out bulk RNA-sequencing analysis to comprehensively characterize three different groups of organoids, 0.25 μM PR3-treated and vehicle-treated patient organoids and control (RC) organoids from three independent differentiation experiments. Consistent with the qRT-PCR gene expression analysis, the results showed a significant downregulation in RHO and other rod phototransduction genes." Here, the authors make it clear that they have performed RNA-seq on three types of samples: PR3-treated patient organoids, vehicle-treated patient organoids, and control organoids (presumably not treated). Yet, in the next sentence, they state "the results showed a significant downregulation in RHO", but they don't state what two of the three conditions are being compared! Although I can assume that the comparison presented in Fig. 6A is between patient vehicle-treated and PR3-treated organoids, this is nowhere explicitly stated in the manuscript.

Thank you for the comment and we will explicitly state various comparisons in the revised version.

(4) There are multiple flaws in the analysis and interpretation of the PR3 treatment results. The authors wrote (lines 289-2945) "We treated long-term cultured 300-days-old, RHO-CNV patient retinal organoids with varying concentrations of PR3 (0.1, 0.25 and 0.5 μM) for one week and assessed the effects on RHO mRNA expression and protein localization. Immunofluorescence staining of PR3-treated organoids displayed a partial rescue of RHO localization with optimal trafficking observed in the 0.25 μM PR3-treated organoids (Figure 5B). None of the organoids showed any evidence of toxicity post-treatment."There are multiple problems here. First, the results are impressionistic and not quantitative. Second, it's not clear that the investigator was blinded with respect to the treatment condition. Third, in the sections presented, the organoids look much more disorganized in the PR3-treated conditions than in the control. In particular, the ONL looks much more poorly formed. Overall, I'd say the organoids looked considerably worse in the 0.25 and 0.5 microM conditions than in the control, but I don't know whether or not the images are representative. Without rigorously quantitative and blinded analysis, it is impossible to draw solid conclusions here. Lastly, the authors state that "none of the organoids showed any evidence of toxicity post-treatment," but do not explain what criteria were used to determine that there was no toxicity.

Thank you for your critical insight. The RHO localization data is qualitative as it is very difficult to accurately quantify rhodopsin trafficking within the cell in the organoid. Thus, for quantitative comparison, we have provided expression level changes. Regarding toxicity, we analyzed the organoids by morphology and TUNEL and did not observe significant difference between the conditions. This closely mimics mouse data on PR3 which suppressed rod function in mice following IP injection without any obvious toxicity.

(5) qPCR-based quantitation of rod gene expression changes in response to PR3 treatment is not well-designed. In lines 294-297 the authors wrote "PR3 drove a significant downregulation of RHO in a dose-dependent manner. Following qRT-PCR analysis, we observed a 2-to-5 log2FC decrease in RHO expression, along with smaller decreases in other rod-specific genes including NR2E3, GNAT1 and PDE6B." I assume these analyses were performed on cDNA derived from whole organoids. There are two problems with this analysis/interpretation. First, a decrease in rod gene expression can be caused by a decrease in the number of rods in the treated organoids (e.g., by cell death) or by a decrease in the expression of rod genes within individual rods. The authors do not distinguish between these two possibilities. Second, as stated above, the percentage of cells that are rods in a given organoid can vary from organoid to organoid. So, to determine whether there is downregulation of rod gene expression, one should ideally perform the qPCR analysis on purified rods.

The reviewer is correct in pointing the potential reasons for reduction in RHO levels following PR3 treatment. Thus, we have provided NRL expression levels in the graph to show that this key rod-specific gene does not change suggesting equivalent number of rod photoreceptor cells. The suggestion of using purified rods is not practical here, as we do not have any way to sort human rods due to the lack of a rod-specific cell surface marker.

(6) In Figure 4B 'RM' panels, the authors show RHO staining around the somata of 'rods' but the inset images suggest that several of these cells lack both NRL and OTX2 staining in their nuclei. All rods should be positive for NRL. Conversely, the same image shows a layer of cells scleral to the cells with putative RHO somal staining which do not show somal staining, and yet they do appear to be positive for NRL and OTX2. What is going on here? The authors need to provide interpretations for these findings.

Since RHO is a cytoplasmic marker and photoreceptor are tightly packed, it is difficult to make a 1:1 comparison to NRL/OTX2 nuclear marker to RHO. Additionally, as the RHO+ cytoplasm moves towards scleral surface, it is expected to pass adjacent to other nuclei. Few of the rods do still have normal Rhodopsin trafficking and it is likely these will not have somal RHO similar to control conditions. We do rarely observe these cells as highlighted by the occasional RHO in IS/OS of RM organoids in the figure. We do agree that the NRL staining in the figure 4B (>D250) is not extremely crisp and we will include an updated figure in the revised version.